

# Plume spreading test case for coastal ocean models

Vera Fofonova[1], Tuomas Kärnä[2], Knut Klingbeil[3], Alexey Androsov[1], Ivan Kuznetsov[1], Dmitry Sidorenko[1], Sergey Danilov[1], Hans Burchard[3], and Karen Helen Wiltshire[1]

[1]Alfred Wegener Institute for Polar and Marine Research. Bremerhaven, Germany.
[2]Finnish Meteorological Institute. Helsinki, Finland.
[3]Leibniz Institute for Baltic Sea Research Warnemünde (IOW), Rostock, Germany.

**Correspondence:** Vera Fofonova (vera.fofonova@awi.de)

**Abstract.**

We present a test case of river plume spreading to evaluate numerical methods used in coastal ocean modeling. It includes an estuary-shelf system whose dynamics combine nonlinear flow regimes with sharp frontal boundaries and linear regimes with cross-shore geostrophic balance. This system is highly sensitive to physical or numerical dissipation and mixing. The main characteristics of the plume dynamics are predicted analytically, but are difficult to reproduce numerically because of numerical mixing present in the models. Our test case reveals the level of numerical mixing as well as the ability of models to reproduce nonlinear processes and frontal zone dynamics. We document numerical solutions for Thetis and FESOM-C models on an unstructured triangular mesh, as well as ones for GETM and FESOM-C models on a quadrilateral mesh. We propose an analysis of simulated plume spreading which may be useful in more general studies of plume dynamics. The major result of our comparative study is that accuracy in reproducing the analytical solution depends less on the type of applied model architecture or numerical grid than it does on the type of advection scheme.

## 1 Introduction

Rivers supply coastal areas with freshwater and both organic and inorganic materials. The correct representation of river mouth dynamics and plume spreading in a numerical coastal ocean model is a prerequisite for accurate simulation of biogeochemical water content and ecosystem dynamics. If we consider river plumes as zones under freshwater influence beginning from the source of freshwater, we would naturally embrace a wide range of processes with different spatial and temporal scales. They would include (but not be limited to) geostrophic currents, frontal processes, and a wide range of mixing processes induced by river momentum, stratified shear, wind and tidal forcing. The expression of these processes as well as river plume behavior in general, depends heavily on local topography at the river mouth, bathymetry detail, discharge characteristics (such as the induced density gradient and discharge rate), and the local Coriolis parameter. These parameters are usually the basis for predicting plume behavior and plume classifications (e.g., Whitehead, 1985; Garvine, 1995, 1987; Yankovsky and Chapman, 1997; Avicola and Huq, 2002, 2003; Chant, 2011; Horner-Devine et al., 2015).

The prototypical plume structure (e.g., Horner-Devine et al., 2015) assumes the presence of a source zone (where initial buoyancy and momentum fluxes are introduced), a near-field, a bulge area, and a coastal current (Fig. 1). These areas are dif-





ferentiated based on the processes which dominate the momentum balance. However, these zones are not mutually independent and should be treated as an interconnected system. Local conditions can prevent the representation of one or another of these plume-structural elements (e.g., Garvine, 1984; Yankovsky and Chapman, 1997; Hetland, 2005; Horner-Devine et al., 2015, 2009). Furthermore buoyant plumes can be categorized separately either as bottom-advected, surface-advected or intermediate (Yankovsky and Chapman, 1997). In this work we will only focus on surface-advected plumes; they are detached from the

bottom and their dynamics are not influenced by near-bottom processes.

According to the review of the subject by Horner-Devine et al. (2015), the near-field zone is a jet-like zone encompassing the mouth area, where river momentum predominates over buoyancy. Here typically lies the so-called 'lift-off' region for surface-advected plumes; across this region, river water loses contact with the bottom, and the interface rises rapidly seaward. The dynamics of the near-field zone suggest intense mixing.

The bulge zone (or mid-field zone) is the area where Earth's rotation begins to predominate, turning the plume down-coast (anticyclonically in the Northern Hemisphere) and creating a gyre. The bulge zone is pronounced in surface advected plumes if river mouths are relatively narrow compared to the Rossby deformation radius and if large mixing sources and ambient currents are absent (e.g., Horner-Devine et al., 2015, 2009; Yankovsky and Chapman, 1997; Garvine, 1995; Avicola and Huq, 2003; Huq, 2009).

Near the coast one portion of the bulge water returns to the gyre while another transforms into the coastal current. This bifurcation area is characterized by predominance of the non-linear terms in the momentum balance, with small effect of horizontal pressure gradient (e.g., Beardsley et al., 1985; Garvine, 1987). The proportion of water returning to the gyre or transforming into the coastal current and the position of the bifurcation area depend on the water flow characteristic angle in the near- and mid-field areas (e.g., Garvine, 1987; Avicola and Huq, 2003; Horner-Devine et al., 2006; Whitehead, 1985). It

should be noted that after a (typically short) time interval of 1-2 inertial periods from the beginning of the plume history, the gyre enters into a gradient-wind balance despite continuing to dilate (Nof and Pichevin, 2001; Horner-Devine et al., 2015, 2009).

Coastal current is a feature typical of all plumes (surface- and bottom-advected as well as intermediate); it represents a buoyancy-driven current in the presence of the Earth's rotation. Being in nearly geostrophic balance, it stays adjacent to the

coast and propagates to the right in the Northern Hemisphere.

Despite the fact that plume behavior has been simulated, observed and reproduced in the laboratory in many configurations (e.g., Whitehead, 1985; Garvine, 1987, 1995; Yankovsky and Chapman, 1997; Fong and Geyer, 2002; Avicola and Huq, 2003; Huq, 2009; de Boer et al., 2009; Liu et al., 2009; Hetland, 2005, 2010; Horner-Devine et al., 2015, 2006; Kärnä, 2020; Chawla et al., 2008; Fischer et al., 2009; Vallaeys et al., 2018; Beardsley and Hart, 1978; Chen et al., 2009), the analysis

of the requirements and limitations helping to reproduce plume behavior in a numerical model is still missing. In particular, spurious numerical mixing in circulation models can destroy stratification and frontal features, and significantly alter the plume dynamics. Spurious numerical mixing can be attributed to the advection schemes (e.g., Burchard and Rennau, 2008; Klingbeil et al., 2014), vertical grid (Griffies et al., 2000; Hofmeister et al., 2010, 2011; Gibson et al., 2017), discretization or time-stepping. Some idealized test cases allow diagnosing spurious mixing (see, e.g., Ilıcak et al., 2012). Also the effect





of numerical mixing on estuarine processes has been demonstrated in several studies (e.g., Kärnä and Baptista, 2016; Ralston et al., 2017; Burchard, 2020). However, the effect of spurious mixing on plume dynamics is still poorly understood.

We have therefore devised a test case that deals with a geometrically simple river-shelf system which has an analytical solution and is very sensitive to numerical and physical mixing. The existing extensive work on plume dynamics allowed us to predict both qualitatively and quantitatively how the plume would behave in the various zones depending on the initial parameters of the system. We have stopped at a river channel oriented perpendicularly to the shelf to ensure that domain geometry is representable with both structured and unstructured meshes. We selected discharge parameters ensuring supercritical flow in the river mouth area (even though the balance near the bottom is fragile in a sense that the numerical mixing can easily cause the penetration of the dense water into the channel). In this case long internal wave disturbances can travel only upstream. Adjusting the configuration further, which included the width of the estuary, discharge rate, density gradient, and Coriolis parameter, we created a system with a very thin surface-advected plume comprising all the classical zones and characterized by a nearly symmetrical (nearshore radius–the distance from the coast to the bulge center–nearly equals offshore one) and pronounced bulge (75 % of the river discharge should stay there). Despite the geometrical simplicity of the test case, the analytically predicted behavior of the plume is hard to reproduce numerically. The described bulge features and mouth dynamics with naturally meandering isopycnals are responsible for the the sensitivity of the test case to any source of mixing - physical, numerical, vertical or horizontal. This feature distinguishes the proposed test-case from other simulations of natural plume systems, most of which are not as sensitive to numerical mixing. We introduced no additional mixing sources (such as wind and tidal forcing) into the proposed test-case (such a step will be easy to accomplish in subsequent set-ups), and used a zero eddy diffusivity coefficient to be able to compare the numerical results with the analytical solution and to have a transparent diagnostic of numerical mixing.

Due to availability of the reference solution and spatial design, the test case can serve multiple purposes: to diagnose how well the numerical solution reproduces the complex multi-scale dynamics of the plume formation and spreading; to test stability and quality of tracer advection schemes (with and without limiters); to determine the level of numerical mixing in simulated flows; and to gauge freshwater mass conservation.

High-order advection schemes (with various limiters) are currently being implemented in coastal models. They are more accurate but also more resource-intensive. It is crucial to understand their limits as well as where and how they can be applied successfully in practice, and the proposed test case is well suited for that. Its advantages include simple preparation and set-up, simple output analysis, short simulation periods, and straightforward interpretation of why plume behaviour can deviate from the analytical solution. Model users wishing to apply models to explore baroclinically dominated flows may also find it useful because it immediately reveals possible gaps in the dynamics under a given set of parameters and limiters, and gives a sense of the fidelity of the model.

We also describe a set of simple and efficient diagnostics of numerical diffusive transport intended to test the performance of tracer advection schemes, limiters, time-stepping and diffusive filters. The brief test of the model performance and its level of numerical diffusion includes only the diagnostic of the offshore spreading of the bulge.





The article provides some new insights into plume dynamics. In particular, the theoretical prediction of the plume behaviour
is derived, explained, simulated and analysed.

The article is organized as follows. The next section describes the modelling setup including information about basic parameters and notation. Then the section with an analytical solution for the test case follows. The next two sections present the numerical results, their analysis and discussion. The last section summarises information for the user. Here, very short summary of the test case, its analytical solution, output requirements and its analysis are given.

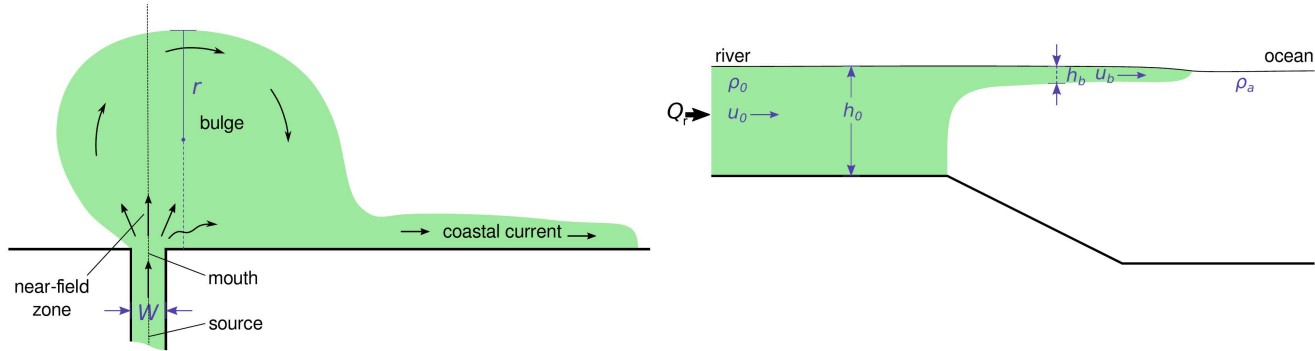

**Figure 1.** The sketch of the river-ocean dynamical system. Left panel: prototypical plume structure. Right panel: vertical cross-section marked by the black dashed line in the left panel.

## 2  Modelling setup

The test case simulates a surface-advected plume with non-trivial near-shore dynamics and all four prototypical zones (Fig.1). To be able to compare the simulated behavior with the analytical solution the eddy diffusivity coefficient is set to zero. There is no forcing except for river discharge. The integration domain is closed except for the river. The system can be considered as a two-layer one for analytical consideration.

The comparison with analytical solution is focused on the position of the lift-off zone, bulge characteristics at a given time (offshore spread — the width of the bulge, and alongshore diameter — the length of the bulge), the depth of the coastal current, its cross-front width, and velocity. The details of required output are summarized in the last section of the manuscript.

### 2.1  The basic notation and parameters

The basic notation and parameters of the test case are presented below. The parameters for the additional set of experiments
used in the Discussion section are given in the brackets.

$W = 0.5$ km is the width of the mouth,

$h_0 = 10$ m is the inflow depth,

$Q_r = 3000(3900)$ m$^3$/s is the river discharge rate,

$f = 1.2 \cdot 10^{-4}$ s$^{-1}$ is the Coriolis parameter,





$h_b$ is the averaged thickness of layer occupied by plume (buoyancy layer) on the shelf,

$H$ is the full depth,

$u_0 \cong 0.6(0.78)$ m/s $(Q_r/(W h_0))$ is the river velocity in the channel in a steady regime,

$u_b$ is the averaged velocity of the layer occupied by plume,

$\rho_r = 1000.65$ kg/m$^3$ is the density of river water,

$\rho_0 = 1023.66$ kg/m$^3$ is the ambient/shelf water density,

$g' = g \frac{\rho_a - \rho_0}{\rho_a} \approx 0.225$ m/s$^2$ is the reduced gravity,

$c_0 = \sqrt{g' h_0} \approx 1.5$ m/s is the reference phase speed,

$r_0 = \frac{c_0}{f} = 12.5$ km is the inflow Rossby radius,

$c_b = \sqrt{g' h_b}$ is the internal gravity wave speed,

$r_b = \frac{c_b}{f}$ is the internal Rossby radius,

$L_0 = \frac{u_0}{f} = 5(6.5)$ km is the inertial radius,

$L_b = \sqrt[4]{\frac{2 Q_r g'}{f^3}} \approx 5.28(5.65)$ km is the internal Rossby radius for the bulge based on the geostrophic depth,

$Fr_0 = \frac{u_0}{c_0} = 0.4(0.52)$ is the initial Froude number,

$T_0 = \frac{2\pi}{f} = 14.54$ h is the inertial period.

**2.2   Setup description**

We consider a steady flow of a brackish water through a narrow channel into a wide, uniformly sloping shelf with relatively dense and initially motionless water. The straight river channel is 10 km long and 0.5 km wide. The shelf zone occupies a rectangular domain 700 km × 500 km (Fig. 1). The river channel divides the shelf coastal line into fragments of 300 km and 400 km to the west and east respectively. The water depth in the channel is 10 m; on the shelf, it increases linearly from 10 m

at the coast to 30 m offshore forming a slope of 0.003; further offshore the depth stays constant at 30 m. The river discharge is set to 3000 m$^3$/s.

The river water is fresh with zero salinity. The shelf water has salinity 30 in practical scale ('practical scale' is omitted below). For the sake of simplicity, temperature is kept constant at 15 $^o$C. In the current work we use mostly monotonic advection schemes (ensured by limiters) and a linear equation of state (therefore, the appearance of small negative salinity is

accepted):

$$\rho(S) = \rho_0 + 0.767 \, \text{kg}^2 \, \text{m}^{-3} \, \text{g}^{-1} (S - 30 \, \text{g} \, \text{kg}^{-1}). \tag{1}$$

The initial value of salinity in the river channel should be equal to river salinity. We recommend to increase the discharge linearly to the reference value of 3000 m$^3$/s during first simulated hour to avoid an initial shock.

The offshore boundaries are impermeable to enable tracing the freshwater volume conservation. The Coriolis parameter is

$1.2 \cdot 10^{-4}$ s$^{-1}$. The simulation time is limited to 35 hours. Although the plume parameters in the test case are not significantly influenced by bottom friction during this time because the plume is confined to the surface, we suggest deactivating bottom friction in simulations. The eddy viscosity coefficient is calculated based on second-order turbulence model ($k - \epsilon$ style), the





horizontal viscosity is set to zero.

We performed the simulations on triangular and quadrilateral meshes with variable resolution. The quadrilateral mesh is a bit coarser in the plume spreading area (Fig. 2). The triangular mesh consists of 76524 triangles and 37900 vertices. The quadrilateral mesh consists of 59706 vertices and 59122 cells.

The 3D reference grid has $k_{\mathrm{max}} = 40$ sigma layers zoomed parabolically towards the surface at $\sigma(0)$:

$$\mid \sigma(k) \mid = \left( \frac{k}{k_{\mathrm{max}}} \right)^2, \qquad \text{for } k = 0, ..., k_{\mathrm{max}}. \qquad (2)$$

Note, the sign of the sigma depends on the code realization and how the z axis is directed.

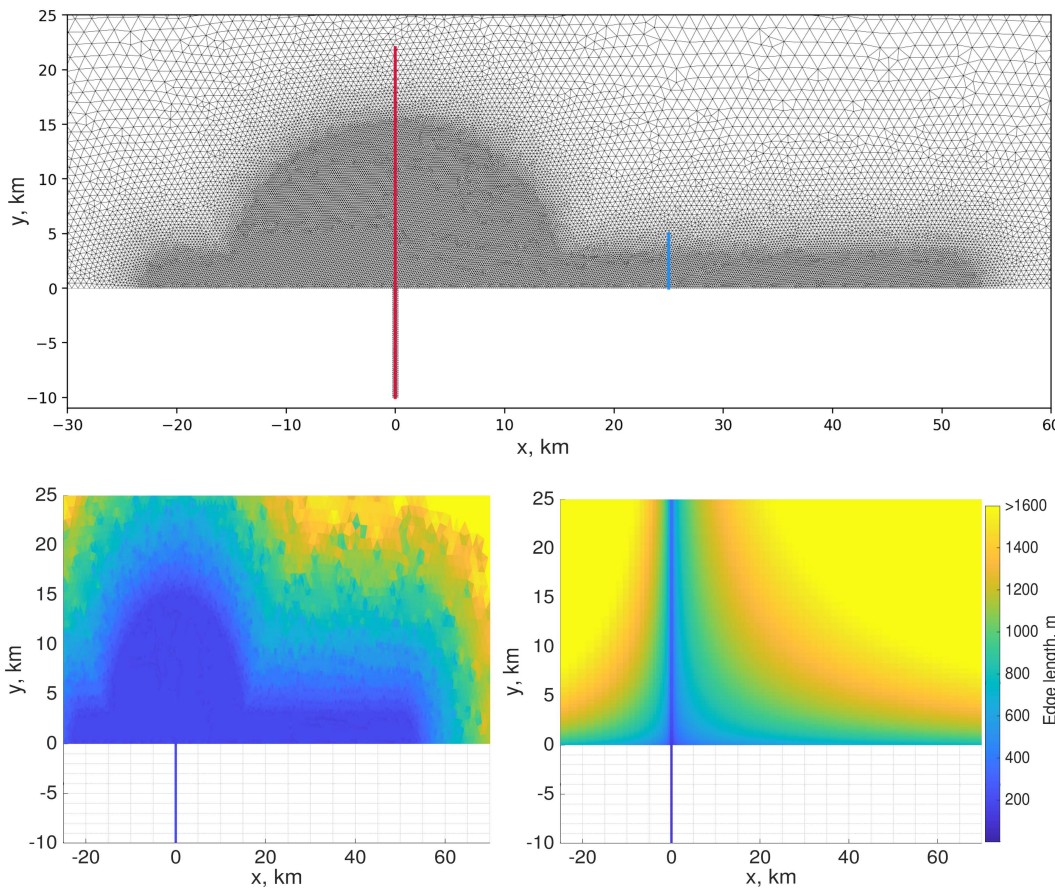

**Figure 2.** Top panel: the triangular mesh with a refinement in the plume spreading area; the lines indicate cross-section positions. Bottom panels: edge length of the triangular and quadrilateral meshes, m, with a zoom in the plume spreading area.



Since our main intention is to learn about the effect of hidden mixing, the experiments mostly differ by the used advection schemes and limiters. Their description is presented in Table 1. The models, advection schemes and limiters are described below.

| Number | Adv. scheme | Turbulence closure for tracer equation | Limiter | Turbulence closure for momentum equation | N of vertical sigma levels with parabolic distr. | Model/grid |
|---|---|---|---|---|---|---|
| 1 (default) | 85% of 3rd order + 15% of 4th order | off | fct1 | k-e | 41 | FESOM-C/tri |
| 2 | 2nd order (upwind) | off | geom. lim | k-e | 41 | Thetis/tri |
| 3 | 85% of 3rd order + 15% of 4th order | off | fct2 | k-e | 41 | FESOM-C/tri |
| 4 | 85% of 3rd order + 15% of 4th order | off | fct3 | k-e | 41 | FESOM-C/tri |
| 5 | 2nd order (Miura) | off | fct1 | k-e | 41 | FESOM-C/tri |
| 6 | 2nd order (upwind) | off | no | k-e | 41 | FESOM-C/tri |
| 7 | 85% of 3rd order + 15% of 4th order | on | fct1 | k-e | 41 | FESOM-C/tri |
| 8 | 85% of 3rd order + 15% of 4th order | off | fct1 | const. vertical eddy visc. coeff., $m^2/s$: a) 3e-4 b) 1e-3 c) 0.1 | 41 | FESOM-C/tri |
| 9 | 85% of 3rd order + 15% of 4th order | off | fct1 | k-e | 21 | FESOM-C/tri |
| 10 | 2nd order (Miura) | off | fct1 | k-e | 21 | FESOM-C/tri |
| 11 | 85% of 3rd order + 15% of 4th order | off | fct1 | k-e | 41 | FESOM-C/quad |
| 12 | 2nd-order (TVD) | off | superbee | k-e | 41 | GETM/quad |
| 13 | 3d order HSIMT (TVD) | off | Sweby's lim. | k-e | 41 | GETM/quad |

**Table 1.** The description of the setups. Blue shading indicates the changes compared to the default run.





Discussion of the viscosity effect is largely based on the additional set of simulations with constant vertical eddy viscosity

coefficient equaled to $2.5 * 10^{-4}$ m²/s and increased discharge equaled to 3900 m³/s (Table A1). (These simulations are not presented in the Results Section.)

## 2.3 Circulation models

### 2.3.1 FESOM-C

FESOM-C is a coastal branch of the global Finite volumE Sea ice Ocean Model (FESOM2) (Danilov et al., 2017; Androsov

et al., 2019; Kuznetsov et al., 2020; Fofonova et al., 2019). FESOM-C solves 3-D primitive equations in the Boussinesq, hydrostatic and traditional approximations for the momentum, continuity, and density constituents, and uses a terrain-following coordinate in the vertical. It has the cell-vertex finite volume discretization (quasi B-grid) and works on any configurations of triangular, quadrangular or hybrid meshes (Androsov et al., 2019; Danilov and Androsov, 2015). The schemes to compute vertical eddy viscosity and diffusivity can be chosen in the frame of the implemented General Ocean Turbulence Model

(Burchard et al., 1999).

FESOM-C splits the fast and slow motions into barotropic and baroclinic subsystems (Lazure and Dumas, 2008). It uses an asynchronous time-stepping, assuming that integration of temperature and salinity is half-step shifted with respect to momentum.

Three schemes have been implemented in the FESOM-C for horizontal advection (Androsov et al., 2019). The first two

are based on linear reconstruction and are therefore second-order. The linear reconstruction upwind scheme and the Miura scheme (Miura, 2007) differ in the implementation of time stepping. The first of them needs the Adams-Bashforth method to be second-order with respect to time. The scheme by Miura reaches this by approximately tracking the volumes advected through the faces of control volumes.

The third approach used in the model is based on the gradient reconstruction (MUSCL-type scheme). The idea of this

approach is to estimate the tracer at control volume faces by a linear reconstruction using the combination of centered and upwind gradients. It represents a combination of the 3rd order and 4th order fluxes in space in a fraction of 85% and 15% respectively. It has the second order in time.

The upwind advection scheme is used together with the third order scheme in vertical, the Miura and MUSCL-type schemes use the 4th order vertical advection.

In this paper the tracer advection schemes are run in a combination with the FCT (flux corrected transport) algorithm. We use three options to constrain the antidiffusive flux which is added to the solution obtained with the positivity-preserving low-order scheme. In the *fct1* option the admissible increments for each scalar point are sought over its horizontal neighbors and in the clusters above and below them, which leaves wide bounds because typically vertical gradients are larger. The *fct2* option is similar to the *fct1* except for the search of vertical bounds which is done locally (tracer values above and below). In the *fct3*

option the vertical bounds are taken into account only if they are narrower than the horizontal ones. In all options the admissible increments are computed with respect to the combination of the low-order solution and the solution at the previous time step.





### 2.3.2 Thetis

Thetis is a 3D hydrostatic circulation model based on Discontinuous Galerkin (DG) finite element formulation (Kärnä et al., 2018; Kärnä, 2020). Thetis uses an unstructured triangular or quad mesh in the horizontal direction, and an extruded prismatic 3D mesh. All prognostic variables are discretized with linear discontinuous elements ($P_1^{DG}$); in 3D the elements are $P_1^{DG}$ in both horizontal and vertical direction. A second order Strong Stability Preserving (SSP) Runge-Kutta time integrator is used to march the equations in time. The 3D dynamics are treated explicitly except vertical diffusion which is treated implicitly; split-implicit mode-splitting technique is used to solve the free surface dynamics. The 3D mesh tracks the position of the free surface. Tracer advection is implemented with upwind fluxes at element interfaces. In addition, a *geometric slope* limiter (Kuzmin, 2010) is a employed as a post-process step to suppress overshoots: if tracer value at a node exceeds the maximum mean value of the neighboring elements, it is marked as an overshoot. The tracer is then redistributed in the element, until none of the nodes violate the extrema conditions, or the element is fully mixed (i.e. all nodes have the same value). The geometric limiters and the SSP time integration method guarantees that the advection scheme is positive definite, i.e. no spurious overshoots are generated. The same advection scheme is applied to both tracers and momentum. The discretization of Thetis dynamical core is described in Kärnä et al. (2018).

The Thetis solver is formally second order in space and time with the exception of vertical diffusion (first order in time) and areas where the slope limiter is active (reducing the scheme to first order). Especially the slope limiters do impose some numerical diffusion to the solution even in the case tracer diffusion operators are omitted.

### 2.3.3 GETM

GETM is an open-source coastal ocean model (Burchard and Bolding, 2002, www.getm.eu). It solves the Reynolds-Averaged Navier-Stokes equations under the Boussinesq approximation together with transport equations for temperature and salinity on a C-staggered structured finite-volume grid. For the present study the non-hydrostatic option by Klingbeil and Burchard (2013) and the temperature equation are not activated, and the potential density is calculated by the linear equation of state (1). The numerics of GETM are similar to other coastal ocean models (Klingbeil et al., 2018). The free surface is integrated in a split-explicit mode-splitting procedure. For the present study the 3D timestep of $60\,\mathrm{s}$ is split into 24 subcycles. In the vertical GETM supports adaptive terrain-following coordinates (Hofmeister et al., 2010; Gräwe et al., 2015), but for the present study fixed sigma coordinates according to (2) are used. Advection of momentum and tracers is carried out by directional splitting with TVD schemes. In the current study the same scheme was used for both advection of momentum and tracer. In order to induce minimum numerical mixing (Klingbeil et al., 2014), in the present study the *superbee* limiter is applied, which is known by its anti-diffusive (anti-viscous) character, in a combination with the 2nd order advection scheme (spatially and temporally). Additionally, the HSIMT 3rd order TVD scheme (2nd order temporally) equipped with *Sweby's* limiter is applied (Wu and Zhu, 2010). If necessary, in individual water columns the vertical advection is automatically iterated to comply with the CFL condition in very thin cells. The turbulent vertical viscosity is calculated by the turbulence module of GOTM (Burchard et al., 1999) in terms of stratification and shear provided by GETM.





## 3 Analytical prediction for the plume behavior


In this section we summarise the qualitative and quantitative predictions of the the plume behavior in the absence of diffusive processes and shortly if they are present in the system (see the respective notes at the end of the subsections) during two first inertial periods.

### 3.1 Mouth area and near-field zone

The mouth area represents a control section for classical hydraulic channel flow (Gill, 1977). The narrow mouth causes rapid shoaling and seaward expansion of the pycnocline, and an acceleration of the upper fresh layer with a significant Froude number gradient, reaching supercritical conditions as freshwater comes out of the river channel. In effect this means that the entire disturbance is swept downstream. Shear increases at the base of an accelerating plume, resulting in very pronounced viscous effects. The presence of the supercritical conditions causes a nearly fully inertial momentum balance (Garvine, 1987).

The acceleration of the plume in the near mouth area leads to a drop in surface pressure following the Bernoulli principle, such that we expect a local drop in sea surface height relative to the channel (e.g., Hetland, 2010). In the limit of zero eddy diffusivity in the tracer equation, for the narrow river mouth and large discharge, the water exchange between the river channel and the shelf is very limited due to constraints imposed by hydraulic control, river momentum (e.g., Gill, 1977; Stommel and Farmer, 1953; Farmer and Armi, 1986; Hetland, 2010; Armi and Farmer, 1986) and the Knudsen (1900) relation. The experiments

with larger discharge –3900 m$^3$/s – suppose that the interface in the mouth area between layers of different densities should reach the bottom (the freshwater should extend all the way to the bottom) given absence of bottom friction and presence of relatively large river velocities (which are larger than the frontier velocities in the lock-exchange experiment corresponding to the given pressure gradient). According to the Armi and Farmer (1986) we are in the case of intermediate barotropic flow (induced by river momentum), when the dense water stays motionless and does not penetrate through the constriction (in our

case it is mouth). Thus we are expecting that the area where the freshwater flow loses contact with the bottom and a *plume* actually forms is situated directly in the mouth area (e.g., MacDonald and Geyer, 2005).

The experiments with a smaller discharge – 3000 m$^3$/s – suppose penetration of the dense water into the river channel only in case when the large viscous effect initiated by the hydraulic jump are neglected at least partly (e.g., by prescribed upper bound for the eddy viscosity coefficient or only background viscosity in numerical solutions). In the inviscid theory of Armi and

Farmer (1986) the barotropic force initiated by the river momentum in this case can be characterised as a transition between moderate to intermediated flow conditions, and penetration of the dense water into the river channel can take place. The viscous effects naturally block the dense water penetration into the river channel, however numerical mixing may provoke it. Even in the absence of explicit diffusion operators there is some mixing in numerical simulations. Numerical mixing is largely attributed to the advection scheme and the limiters built in it. It can lead to dense water penetration into the river channel and the appearance

of new salinity classes in the river channel. For any tracer c, the total (advective plus diffusive) tracer flux per unit area of a transect or isohaline can be defined as: $F^c = u_n c - K_n \partial_n c$ where $u_n$ is the outgoing normal velocity, $\partial_n c$ is the tracer gradient normal to the surface, and $K_n$ is diffusivity in the direction normal to the surface. Let us take the mouth transect directly





upstream from supercritical conditions. It is clear that the presence of a large enough numerical mixing can provoke the dense water penetration into the river channel. The density intrusion in this case will have a hydraulically controlled, blunt-faced

profile (e.g., Jirka and Arita, 1987). If we apply the written above equation to the isohaline of the freshwater layer and activate vertical eddy diffusivity, we will also get dense water penetrating into the system, with the hydraulic control setting the upper limit of exchange as well as a more complex interface (e.g., MacCready and Geyer, 2001). We will return to this topic in the Discussion.

### 3.2   Bulge

Here we revisit the definition of a bulge according to the review by Horner-Devine et al. (2015) as a continuation of the near-field zone in which Earth's rotation in the momentum balance begins to predominate, turning the plume to the right (in the Northern Hemisphere) and creating a gyre in thermal wind balance. Note that the near-field-zone, bulge and coastal current have a very complex dynamic structure, such that the consideration of various discharge characteristics is needed to clearly describe it. We prefer to avoid this by focusing on the resulting plume-spreading characteristics: maximum offshore plume spreading

position, internal bulge radius (nearshore radius) and the along-shore bulge length. Avicola and Huq (2003) had shown that on average the bulge along-shore spread is longer than its offshore spread, and that this ellipticity is constant through time and equal to $\sim 1.3$. Thus it suffices for us to locate the maximum of bulge offshore spreading and let the along-shore scale be a control point.

We know that when the channel is at a right angle to the coast the bulge grows continuously over time because its increasing

size creates a balance between the momentum flux associated with the downstream current and the compensating Coriolis force associated with the migration of the gyre center away from the coast (e.g., Fong et al., 1997; Nof and Pichevin, 2001; Horner-Devine et al., 2006, 2009). But the expansion of the bulge leads to radial and advective acceleration terms that are two orders of magnitude smaller than the terms of the gradient-wind balance. Accordingly, the gradient-wind balance is expected to apply even for a growing bulge as long as its radius is sufficiently large (Horner-Devine et al., 2006, 2009). Recent studies

have shown that the bulge offshore radius and displacement of bulge center from the wall (nearshore radius) can be scaled respectively with the internal Rossby radius ($L_0$) and inertial radius ($L_b$), both of which are constant over time (Horner-Devine et al., 2009). In our case $L^* = \frac{L_0}{L_b} \approx 1$, so the river flow is one of the main factors pushing the bulge offshore, the anticyclonic circulation is nearly symmetric (in terms of nearshore and offshore radii); and the bulge is prone to instability (Horner-Devine et al., 2009). Thus the net shore-directed Coriolis force is small, the angle of incidence of the recirculating bulge flow that it

makes with the coast is greater than $90°$ (flow directed back upstream) (Whitehead, 1985; Horner-Devine et al., 2009), and the majority of the impinging fluid is directed back into the bulge. This feature largely makes the bulge size sensitive to any source of mixing in the model. The laboratory and theoretical studies mentioned above have found that the plume starts turning to the right in the Northern Hemisphere approximately at one-fourth of the rotation period at a radius of about $L_0$ and reaches thermal wind balance after one to two rotation periods.





To obtain an estimate for the offshore spreading of the bulge in the absence of diffusive processes, we re-visit the equations provided by Yankovsky and Chapman (1997), and propose some modifications. We consider flow in the bulge as being in cyclostrophic balance as described by the following momentum equation:

$$-\frac{u_c^2}{r} - fu_c = -g'\frac{\partial h_b}{\partial r}, \tag{3}$$

where $u_c$ is the azimuthal cyclostrophic velocity at the radial distance $r$ from the anticyclone center, $h_b$ is the thickness of the

buoyant layer.

Because of a purely surface-advected plume at the shelf, we assume no interaction of the plume with the bottom and a plume thickness of $h_b$ which changes little from the mouth area to the center of the anticylonic bulge, and gradually decreases to zero along the outer edge. This means that $\frac{\partial h_b}{\partial r}$ is equal to zero along the streamline from the mouth to the anticyclonic center, and it can be expressed as $\frac{-h_c}{r}$ from the bulge center to its offshore edge, where $h_c$ is the depth of the bulge center and $r$ is offshore

radius of the bulge. Returning to (3) we get:

$$r = \frac{-(g'h_c + u_c{}^2)}{fu_c}. \tag{4}$$

To get $u_c$, the Bernoulli function for the buoyant layer (Gill, 1982) can be applied:

$$B = g'h_b + \frac{u_b^2}{2}. \tag{5}$$

*In the absence of diffusion* $B$ should be constant along the streamline. In Yankovsky and Chapman (1997), inflow is con-

nected to the outer edge and $h_c = h_0$. This poses two major problems: (i) $h_0$ and $h_c$ can be significantly different in a purely surface-advected plume as (in our case) a lift-off point is situated immediately at the mouth or even upstream and is accomplished by the hydraulic jump in the mouth; (ii) the bulge continuously grows over time.

As mentioned above, the gradient-wind balance is expected to apply even to a growing bulge as long as its radius is sufficiently large (Horner-Devine et al., 2006, 2009). So (ii) can be addressed as follows: we determine the radius of the bulge when

the plume forms even though at that point the coastal current is already in place and the bulge is in thermal wind balance. Our particular test-case places no focus on a slow mode of bulge growth (as covered in Nof and Pichevin (2001)); our task is rather to obtain a short-term prediction once the bulge has reached thermal wind balance. Point (i) can be addressed by introducing the so-called geostrophic depth, $h_g$, or the depth of the plume in the near-field zone within critical conditions and taking into account that the diffusive processes are absent in our consideration. It is well-known that the inflow momentum is the most

important factor defining the position of the bulge center (e.g., Horner-Devine et al., 2006), and that the depth of the bulge center becomes proportional to the geostrophic depth as soon as the bulge attains the thermal wind balance (e.g., Avicola and Huq, 2003; Yankovsky and Chapman, 1997). The equation for $h_g$ above is based on consideration of a two-layer Margules front system that has a quiescent lower layer and an upper layer in thermal wind balance or in the geostrophic cross-shore momentum balance (valid for a coastal current, see below) with uniform vertical shear of the alongshore plume velocity. Then





we assume that the *entire* buoyancy inflow transport ($=Q_r$) accumulates in the frontal zone (e.g., Yankovsky and Chapman, 1997; Fong and Geyer, 2002). Finally, we get:

$$h_c = h_g = \sqrt{\frac{2W u_0 h_0 f}{g'}} = 1.8(2)\,m. \tag{6}$$

So, we can determine the properties of the bulge when the bulge center is located at this level. Naturally, it takes place only at the beginning of the plume evolution. The bulge continuously grows not only offshore, but also in depth, however, at a slower

rate. The whole discharge accumulates in the bulge only for a very short period of time prior to the appearance of the coastal current. The front is expanding at approximately the surface gravity wave speed within the layer $h_g$; so we connect the outer edge to these flow conditions:

$$\frac{3}{2} g' h_g = \frac{3}{\sqrt{2}} \sqrt{W u_0 h_0 f g'} = \frac{u_c^2}{2} \tag{7}$$

$$u_c = -\sqrt[4]{18 W u_0 h_0 f g'} = -\sqrt[4]{18 Q_r f g'} \approx -1.1(1.17)\,\text{m/s} \tag{8}$$

$$r = \frac{4}{\sqrt{3}} \frac{\sqrt{g' h_g}}{f} \approx 12.2(12.9)\text{km} \tag{9}$$

This radius $r$ in Equation (9) represents an offshore radius of the bulge in our test-case as soon as the gyre is in thermal wind balance and its center is about $h_g$ thick. Based on $L^*$ value calculated above, the nearshore radius should be close to offshore

radius. This can be expected after one to two rotational periods based on laboratory experiments and simple calculations from the internal Rossby radius, $h_g$ or $h_c$ and associated surface gravity wave speed (e.g., Hetland and MacDonald, 2008; Wright and Coleman, 1971; Hetland, 2010). The predicted radius is consistent with laboratory experiments published by Avicola and Huq (2003) and Horner-Devine et al. (2006) after approximately one to two rotational periods as soon as the gyre is in thermal wind balance. However, these experiments predict for such a radius (relatively to the Rossby radius) at least one-and-a half

times (Horner-Devine et al., 2006) or even twice (Avicola and Huq, 2003) as much deepening at the center. (Horner-Devine et al. (2006) related their findings of a smaller central depth to different measurement techniques.) Defining the bulge depth based on reference buoyancy (20% of the inflow buoyancy) instead of the maximum vertical gradient, one obtains greater deepening. Deepening of the gyre center is a relatively slow process and is usually quasi-stationary after several rotational periods. Deepening at the center is largely attributed to mixing and dilution processes at the plume base and the analytical

solution does not consider the influence of diffusive processes. In our simulations we are omitting the physical diffusivity (eddy diffusivity is set to zero in the tracer equation), in order to reproduce the analytical estimations about the bulge offshore spreading. As for the position of the bulge center relative to the $x$-axis, re-circulation of the discharged water can take place only to the right of the river mouth. On the other hand we have defined the bulge in such a way that a part of it is found to the





left of the source. Baroclinic instability can also lead to rotated structures in the area of interest due to relatively fast radial (as

compared to vertical) bulge growth (e.g., Avicola and Huq, 2003). We therefore are not going to find the exact bulge center position defining it only on the $x$-axis as the site of maximum offshore spreading of the bulge.

Note, that in our idealized conditions the bulge becomes nearly symmetrical and tends toward instability, which suggests a solution sensitive to mixing. Any additional diffusion in the bulge zone will directly reduce the bulge external radius, displace its center, and change the angle at which the bulge characteristics impinge upon the coastal zone. A small isohaline area requires

greater mixing than a large one to maintain the same total freshwater flux across the isohaline (Hetland, 2005; Burchard, 2020). So either the bulge tends to be less restricted offshore and in parallel deeper or/and the bulge tends to be less restricted offshore together with reduced discharge rate associated with the bulge (the bulge will be sliced off and impinge angel will be changed).

### 3.3   Coastal current

In this subsection we are going to derive the coastal current characteristics, in particular the bounds for the coastal current

depth near the wall, the bounds for the near wall speed and the coastal current offshore spread. Typically, the coastal current has a quasi-triangle profile in the offshore cross-section, so when we are talking about near the wall depth and speed we mean maximum depth and speed at each offshore cross-section. Below we will omit 'near the wall' for simplicity.

We can calculate the discharge attributed to the coastal current, $Q_{cc}$, based on the current bulge vorticity (e.g., Nof and Pichevin, 2001):

$$\alpha = \frac{-2u_c}{f \cdot r} \approx 1.5(1.51) \tag{10}$$

$$\frac{Q_{cc}}{Q_r} = \frac{1}{1 + 2\alpha} \approx 0.25(0.248) \tag{11}$$

Based on the obtained $Q_{cc}$, we arrive at a minimum freshwater layer thickness to be expected for the coastal current:

$$h_{min} = \sqrt{\frac{2Q_{cc}f}{g'}} \approx 0.9(1)\,m \tag{12}$$

Naturally, the geostrophic depth $h_g$ calculated above gives us the maximum freshwater layer thickness, $h_{max}$, to be expected in the coastal current, if the total discharge from the river goes for some reasons to the coastal current. We have known already that in our case a large portion of the freshwater stays in the bulge. However, when $L^*$ is large, the bulge may be unstable, separating from and re-attaching to the wall and causing a pulsed flow of the coastal current (Horner-Devine et al., 2006). Furthermore when the coastal current forms, a different portion of the freshwater may go with it depending on the time moment.

The geostrophic depth therefore provides a good estimate of the maximum depth of the layer influenced by the coastal current.

The coastal current propagates at a speed given by $c_n = \sqrt{g' \cdot h}$, the propagation speed should therefore be between 0.45 (0.47) and 0.64 (0.67) m/s taking in consideration $h_{min}$ and $h_{max}$ respectively. Based on these values we derive a local Rossby radius equal to 3.75 (3.9)-5.3 (5.6) km. We can expect that the coastal current will occupy this width in the nose zone at the





beginning of the plume history. Predicting the width of the coastal current in the upstream area between the source and the
nose is not a trivial task (a problem already identified by Garvine (1995)). However, it is clear that the position of the bulge
center should largely determine the coastal current maximum offshore width in the considered time frame. We already know
that a nearshore radius of the bulge is of about 12 km after one to two rotational periods, which is at about two internal Rossby
radii. This result is in agreement with the laboratory experiments (Avicola and Huq, 2003; Horner-Devine et al., 2006) where
the width may be up to two local Rossby radii after one to two rotational periods. To summarize we can provide a relatively
wide window for the expected coastal current offshore width, it can vary between 12 km near the source zone for both runs to
3.75 (3.9) km in the nose zone.

Note, that in the presence of non-zero eddy diffusivity, we can expect a larger amount of the discharge to enter the coastal
current, because the bulge in the non-diffusive case is nearly symmetrical (in the sense of internal - near coast- and offshore
radii) and reaches maximum offshore extension. Therefore, in principle, the coastal current discharge could be used as an
indicator for numerical diffusivity. Such an approach is not used here, it would require considering many rotational periods for
a precise estimation.

## 4    Diagnostic of the numerical diffusivity

The amount of dense water that penetrates into the river channel, the coastal current discharge rate and offshore spread as well
as asymmetry or the characteristic impingement angle of the bulge can all be considered as indirect measures of numerical
mixing. However, each of these measures requires additional analysis and has some limitations.

We base our analysis on isohalines and salinity classes following the work by Hetland (2005). Using the balance equation
for salinity and mass conservation law we obtain a budget equation for the salinity integrated over all salinities between the
river salinity $S_r$ and the salinity of the isohaline, $S$:

$$\frac{\partial}{\partial t} \iiint\limits_{S_r \leqslant s \leqslant S} (S - s)\, dV = (S - S_r) \cdot Q_r + \iint\limits_{s=S} \mathbf{f_{diff}} \cdot \mathbf{n}\, d\sigma, \tag{13}$$

where $\mathbf{f_{diff}}$ is the diffusive salinity flux vector through the isohaline $S$, and $\mathbf{n}$ is the outward normal unit vector on the isohaline
$S$. Neglecting physical diffusion and assuming zero river salinity ($S_r = 0$), the diahaline flux is related to numerical mixing,
so that the numerically induced total salinity discharge through the isohalines $S$ can be calculated as:

$$F^s(S) = \iint\limits_{s=S} \mathbf{f_{num}} \cdot \mathbf{n}\, d\sigma = \frac{\partial}{\partial t} \left( \iiint\limits_{0 \leqslant s \leqslant S_i} (S - s)\, dV \right) - S \cdot Q_r. \tag{14}$$

Note, that numerical mixing is largely related to hidden numerical diffusion, but may also include antidiffusive effects. For
further analysis, we divide (14) by $S$, which gives the diahaline diffusive freshwater discharge (Hetland, 2005)(related to
numerical mixing):

$$F(S) = \frac{F^s(S)}{S}. \tag{15}$$





By dividing the discharges $F^s(S)$ and $F(S)$ by the isohaline area, $A(S)$, the respective average diahaline fluxes are obtained:

$$f^s(S) = \frac{F^s(S)}{A(S)}, \quad f(S) = \frac{F(S)}{A(S)}. \tag{16}$$

We further define the diahaline velocity as

$$w^{\mathrm{dia}}(S) = \frac{\partial f^s(S)}{\partial S} \tag{17}$$

and note that only under stationary conditions with $F^s(S) = SQ_r$ both $w^{\mathrm{dia}}(S)$ and $f(S)$ are identical.

For the analysis of the different models results, we will be using time averaged transports and fluxes:

$$\bar{F}^s(S) = \langle F^s(S) \rangle, \ \bar{F}(S) = \langle F(S) \rangle, \ \bar{f}^s(S) = \frac{\langle F^s(S) \rangle}{\langle A(S) \rangle}, \ \bar{f}(S) = \frac{\langle F(S) \rangle}{\langle A(S) \rangle}, \ \bar{w}^{\mathrm{dia}}(S) = \frac{\partial \bar{f}^s(S)}{\partial S} \tag{18}$$

where $\langle \cdot \rangle$ denotes a time average.

For the diagnostics of numerical mixing presented above the salinity range is divided in isohaline classes and the volume of each class is calculated. These volumes are useful diagnostics, as soon as numerical mixing creates the volumes between the first and last salinity classes.

## 5   Results

In describing the results we primarily focus on the first two simulations from Table 1. The differences in the dynamics of these runs facilitate interpretation of other runs. Additional simulations (see Table 1) have been conducted to illustrate the plume dynamics sensitivity to certain parameters. Runs **11**, **12** and **13** are performed on quadrilateral meshes with FESOM-C and GETM, the description of their performance is at the end of the current section. When comparing runs **11, 12** and **13** with others one should keep in mind that the resolution of the rectangular and triangular grids are not identical (Fig. 2). Therefore
the results of these runs are presented separately (except for analysis of numerical diffusion) despite the fact that they are discussed in the same frame.

Table 2 contains information about predicted characteristics of the plume behaviour. To summarize, we are expecting that the bulge offshore spread reaches not less than 24 km after second inertial period accumulating at about 75% of the total freshwater runoff, and that its surface is fresh, and that the coastal current transports only about 25% of the total freshwater runoff. We
stress that these characteristics are, of course, not independent and the offshore spread of the bulge can be treated generally as a final indicator of the model performance if the surface of the bulge is fresh.

Figure 3 compares the surface salinity, velocity and elevation for runs **1** and **2** after the first and the second inertial periods, i.e. after 20 h and 35 h. It illustrates that in both simulations the plume starts turning right after a quarter of the rotational period and at a distance of one inertial radius ($\sim 5$ km). Also, in both simulations the coastal current has began to form after
one rotational period. Nevertheless, the differences between simulations, regarding representation of the bulge and the coastal current dynamics, are substantial already after the first and become even larger after the second inertial period. The main differences between two runs are summarized in Table 2.



**Figure 3.** Results of the runs 1 (left panel) and 2 (right panel) after 20h and 35h: a) Surface salinity, in practical scale, at 20h; b) Surface salinity, in practical scale, at 35h; c) Surface velocity, m/s, at 35h; d) Surface elevation, m, at 35h.

| Run | 1 | | 2 | | Theoretical prediction (laboratory studies) |
|---|---|---|---|---|---|
| Time | 20h | 35h | 20h | 35h | **20 - 35h** |
| **Bulge** | | | | | |
| **Bulge maximum offshore spreading, km** | 16.5 (18.5) | 19.9 (24.6) | 14.5 (19.2) | 16.2 (22.7) | **24** |
| **Bulge length, km** | 24 | 32 | 25 | 29.5 | **31** |
| **Ratio length/width** | 1.45 | 1.6 | 1.72 | 1.77 | **1.3** |
| **Coastal current at cross-section** | | | | | |
| **Buoyancy layer maximum depth, m** | 0.65 | 1.8 | 1 | 1.8 | **0.9 - 1.8** |
| **Mean velocity (buoyancy layer), m/s** | 0.62 | 0.42 | 0.67 | 0.67 | **0.45 - 0.64** |
| **Qcc, m³/s (%)** | 411 (14) | 981 (33) | 868 (29) | 1554 (52) | **~750 (25)** |
| **Coastal current** | | | | | |
| **Maximum width, km** | 5 | 7 | 3.5 | 4.8 | **3.75 - 12** |

**Table 2.** Summary of the results after 20h and 35h of the runs **1** and **2**. The mean value of the coastal current discharge at different time moments should be compared with the analytical solution. The red numbers demonstrate the bulge offshore spread (width) based on simulated length divided by 1.3, where 1.3 is the ratio between the length and width obtained in laboratory study.

## 5.1 Mouth area/near field zone

In agreement with observational studies (e.g., Wright and Coleman, 1971; MacDonald and Geyer, 2004) in both runs the river
water leaves the narrow estuary and rapidly shoals over a distance of a few channel widths. The velocities there reach the initial surface gravity wave speed of $\sim 1.5$ m/s and more, which is also expected (e.g., Hetland and MacDonald, 2008; Wright and Coleman, 1971; Hetland, 2010). Also, in both runs the near field area is characterized by the presence of supercritical conditions and a hydraulic jump (see Fig. 4 and Fig. 6).



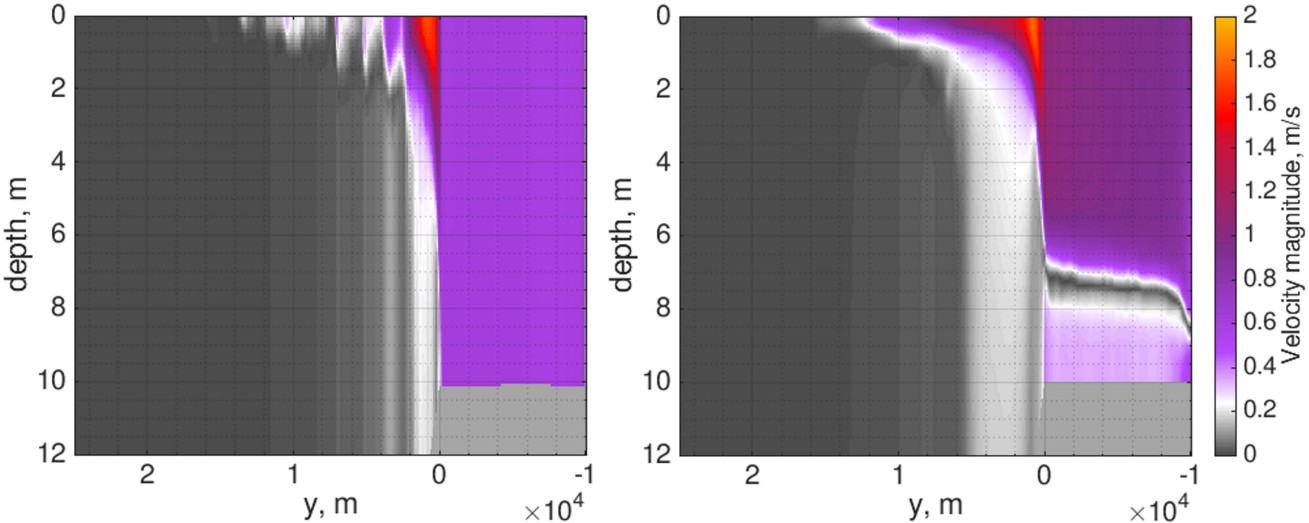

**Figure 4.** The magnitude of the velocity at the mouth-cross section based on runs **1** (left panel) and **2** (right) at 20h time moment.

In run **2** the dense water penetrates into the river channel and depicts a hydraulically controlled blunt-faced profile. We argue

that it is in a large extent caused by the flux limiting scheme which should be most active in this area, or by a low order advection

scheme. Indeed, the limiters are relatively diffusive horizontally in run **2** (as compared to *fct1* and *fct2* options in FESOM-C)

and the advection scheme of the second order has been used. To justify our argument we conducted run **4** with more diffusive

limiter definition (*fct3*, see Modelling Setup section for description of limiters) and run **5** with the second order advection

scheme and relatively low-diffusive limiter option (*fct1*). Figure 5, showing the results after the second inertial period, confirms

that the blunt-faced intrusion profile is simulated in both sensitivity runs. Naturally, both the low order advection scheme and

limiters work toward a more diffusive solution. Interestingly, as it will be shown later, run **2** is characterized by small diahaline

diffusivities for relatively higher salinities in the area of the hydraulic jump as compared to run **1** (Fig. 6) despite the fact that

vertical advection scheme of run **1** is of higher order compared to that of run **2** and also the surface salinity in run **2** is slightly

larger compared to run **1** there (Fig. 3).

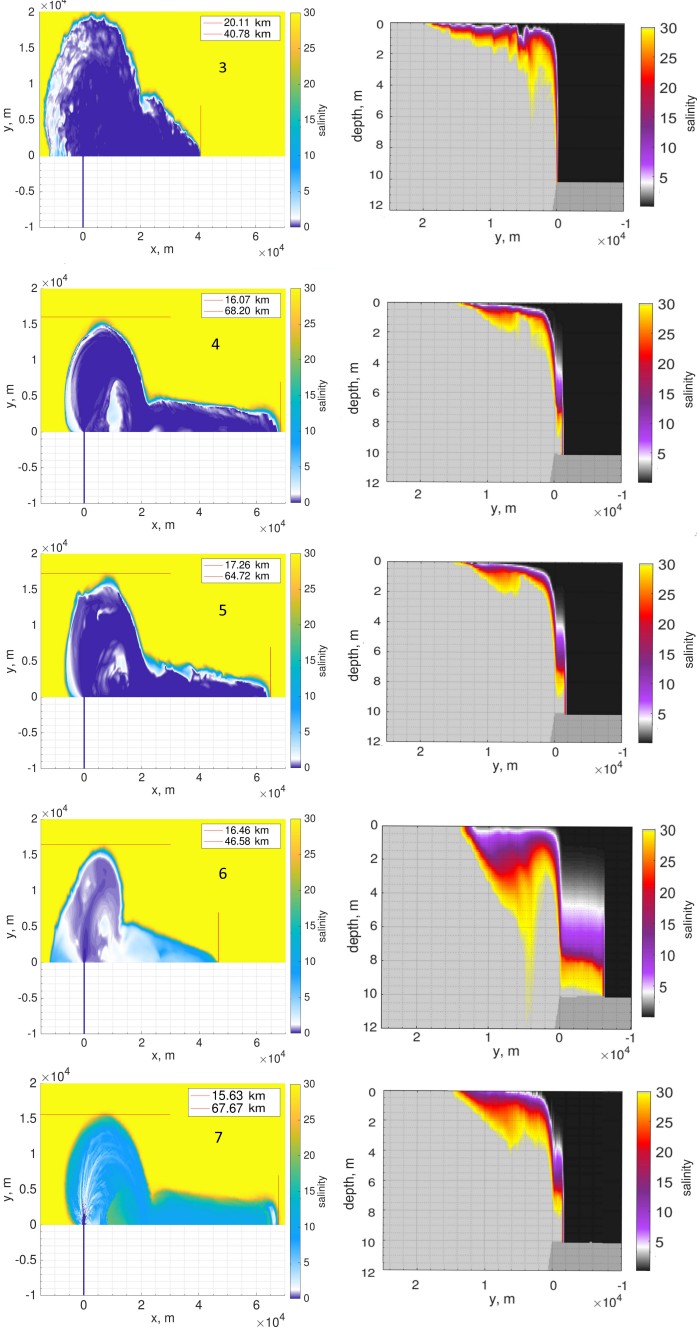

**Figure 5.** The surface salinity (left panel) and salinity at the mouth cross-section (right panel), in practical scale, at 35h in different runs indicated by the number (see Table 1). The bulge offshore spreading and the coastal current propagation are shown by red lines.





Note that in the initialization phase before the river flow reaches the mouth area, a typical lock-exchange experiment dynamical picture emerges in both simulations (initially, we have filled the river channel with river water). However, it vanishes as soon as the river flux reaches the mouth.

We diagnose the internal hydraulics of the model runs by computing the Froude numbers of the plume layer (Fig. 6). The Froude number is computed as $\frac{u_b}{\sqrt{h_b \cdot g'}}$ (where $h_b$ is the plume layer thickness, $u_b$ is the mean velocity within this layer and $g'$ is
the reduced gravity). The layer thickness is defined by the position of the maximum salinity gradient or maximum stress in case of runs **1** and **2** respectively. Due to the different discretization types, FESOM-C velocity fields tend to be noisy on triangular grids (this problem is well known and can be solved by, for example, introducing biharmonic operator, which is omitted for the current study for transparency), whereas in Thetis the tracer fields contain some noise. Nonetheless, the plume layer border in both models follows the ∼6 isohaline. Due to entrainment and numerical diffusion the layer defined by the gradient is thinner
in the area of hydraulic jump, and therefore the values presented should not be treated as absolutely accurate. Hence, calculated Froude numbers can be used to trace the dynamics, but they can deviate and are generally larger than the real Froude numbers within area of supercritical conditions.

The locations of the maximum Froude numbers differ between run **1** and run **2** (see Fig. 6). The difference can be traced in the velocity disturbances underneath the plume layer (Fig. 4) which is generated by the eddy viscosity reacting to the shear
stress at the near surface and the latter is induced by a hydraulic jump. Also in the area directly downstream of where the maximum Froude numbers occur the pronounced salinity finger appears. In case of run **1** the area with the largest velocities (more than initial surface gravity wave speed) is more localized in offshore direction and the maximum Froude numbers are found directly downstream the mouth area.







**Figure 6.** The salinity patterns, in practical scale, at cross-sections based on runs **1** (left panel) and **2** (right panel). The blue line shows the thickness of the plume layer based on maximum vertical gradient of the horizontal velocity/salinity fields. The dark blue circles identify the approximate Froude numbers of the plume layer (do not equal the true Froude numbers).





## 5.2 Bulge and coastal current

Once the bulge (in idealized conditions) is nearly symmetric and tends towards instability, it becomes sensitive to any source of mixing: horizontal or vertical, physical or numerical. Therefore the usage of different advection schemes, limiters, and time-stepping results in different dynamics.

The ratio between the length (across shelf spread) and width (offshore spread) of the bulge is another parameter, which indicates the presence of numerical mixing in the system. In all triangular-mesh configurations including run **1** the ratio is a

too large compared to the expected number (Table 2). Interestingly, the along shore lengths of the bulges in run **1** and run **2** are nearly within the range of analytically predicted values. In run **2** and all others, where an advection scheme of second order or relatively diffusive limiters are used (i.e. runs **4, 5, 6, 10**), the bulge is largely sliced off by the coastal wall. This effect only minor presents in run **1** (see Fig. 3 and Fig. 5).

Further details of the bulge structure can be derived from the $v$-component of the horizontal velocity. Figure 7 shows the

surface $v$-component against $x$-position at a fixed $y$, which equals the internal Rossby radius ($\sim 5.3$ km, based on geostrophic depth). Although the line of a fixed $y$ does not cross the bulge center, its approximate $x$-position can be still identified from Figure 7. Compared to other runs on triangular grid run **1** depicts the largest spread of the bulge to the left from the mouth area. Consequently, the bulge center is most displaced to the left there and is also located further from the coastline than in run **2**.

In run **2** the bulge is less symmetric as expressed by the internal and external offshore radius and, consequently, the freshwater

discharge of the coastal current there is nearly two times larger as compared to run **1**. Here we should say that the position of the second cross-section in Figure 2, which is at the coastal current, is quite far from the mouth area. Therefore, while the front of the coastal current after 20h in run **2** already reached the cross-section, only the nose area of the coastal current in run **1** reaches there after that time. This explains the difference in numbers pertaining to the coastal current at 20 h and 35 h. In average, the coastal current in run **1** and run **2** transports about 25% and 40% respectively of the initial freshwater discharge

during the time period under consideration.



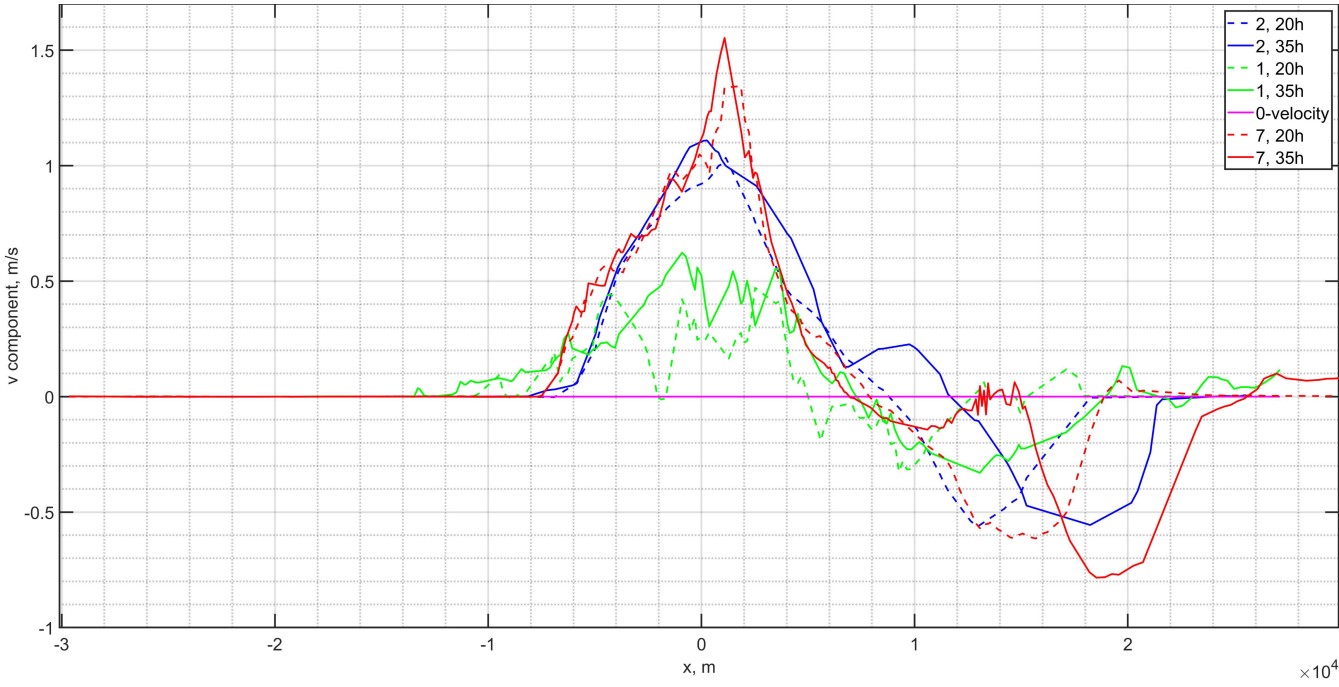

**Figure 7.** The $v$-component of the horizontal velocity at the fixed $y$-axis position equal to the internal Rossby radius, $\sim 5.3$ km, for different runs at 20h and 35h.

The position of the front of the coastal current and the bulge offshore radius can also provide a qualitative estimate of the level of numerical diffusion. The large coastal current discharge (far spreading of the coastal current) and small offshore restriction of the bulge compared to the analytical solution indicates the presence of numerical diffusion in the system (Fig. 5). Note, that the level of numerical diffusion can be even larger if the same small offshore restriction of the bulge is in parallel

with the weakly developed coastal current. In such a case the bulge and the coastal current are excessively thick.

Among triangular discretizations **runs 1, 3** are characterized by the larger bulge with a fresh surface and slower and wider coastal current, which transports less freshwater compared to other runs, and is the closest to analytical solution. Run 3 has slightly different fct limiting details (it uses $fct2$ option, see Setup Section for the details). Also in run **1**, and in some additional runs the velocity and elevation fields (see e.g. Fig. 3 and Fig. 5) depict the presence of the physical instability at frontal zones.

However, the elevation fields there also depict some noise in the areas adjacent to the plume boundaries (Fig. 3). The reason behind this noise are the spurious inertial oscillations which are present on triangular meshes in FESOM-C. Due to the absence of the tracer diffusivity, and special designed filters (e.g., biharmonic filter) and expected low level of numerical diffusion in run **1** such oscillations are not sufficiently damped there and are present in the simulated patterns.





## 5.3 Performance on rectangular grid

Here we consider runs **11**, **12** and **13**. Run **11** has the same set of options as run **1**, but is performed on rectangular mesh (Table 1). As expected, on a rectangular mesh in run **11** there are no inertial oscillations which appear on the triangular mesh due to the discretization type of FESOM-C. The noise occurs only in the places where the grid resolution becomes sharply coarse (see Fig. 2, Fig. 8 b). The plume in run **11** spreads nearly to the same position as in case of run **1**. Due to the coarser grid, the nose of the coastal current appears a bit further compared to run **1**. The surface, occupied by the plume, is nearly fresh everywhere

in run **11** and there is also no penetration of the dense water into the river channel (Fig. 8 a, c).





**Figure 8.** The results of run **11** (left panel), run **12** (middle panel) and run **13** (right panel) at 35h performed on the rectangular grid: a) Surface salinity, in practical scale; b) Elevation, m; c) Salinity pattern at the mouth cross-section, in practical scale; d) Magnitude of the velocity at the mouth cross-section, m/s;





run **12**, which is characterised by the anti-diffusive limiter *superbee*, shows better results than the bulk of the runs presented on triangular grid: the offshore spread of the bulge reaches 17 km, the surface layer occupied by plume is generally fresh. Runs **11, 12** simulate the ratios between alongshore- and offshore- spread of the bulge, which are close to the analytically predicted value of 1.3 and are equal to ∼1.4 and ∼ 1.5 respectively: the bulges are sliced off by the coastal wall in a lesser extent.

Run **13** is characterised by the same offshore plume spread as run **12**. However, the plume surface layer is characterised by the salinity more than 5. It means that run **13** is more diffusive in vertical compared to runs **11, 12**. We can also see that in Figure 8c, d. As can be expected, the coastal current propagates further compared to runs **11, 12**. In run **13** there is also a penetration of the dense water into the river channel, however, it has slightly different interface profile compared to run **12**. The profile is close to the typical lock-exchange one (Fig. 8c, d). It signalises about the advection scheme of higher order in a horizontal

direction in run **13** compared to run **12**, also it means that used limiter (*Sweby's* limiter) does not reduce significantly the order of advection scheme horizontally. The advection scheme of a third order in case of run **13** is also responsible for relatively large offshore spreading of the bulge.

## 5.4 Numerical diffusivity and model performance

In the model runs where the physical diffusivity is set to zero any source of mixing of tracers is of numerical origin. In order

to contrast it to a physical diffusivity a run **7** with the physical diffusivity switched on (Table 1) will be also presented.

For simplicity for all runs analysis, we replace the isohaline area by its surface projection: the isohaline area is represented by the sum of control areas of vertexes or elements depending on the discretization, where particular salinity is present at any depth. With the exception of potentially very complex isohaline configurations in the near-field zone, this is a reasonable assumption in as much as the diffusive processes mediate offshore spreading of the plume and its final deviation from the

analytical solution where there is no additional forcing in the system (this issue is also presented in the Discussion). We should just mention that the level of numerical mixing would be slightly overestimated through that for less diffusive solutions that tolerate meandering isohalines (Fig. 4, 6). Note that the true isohaline areas can readily taken into account if a more accurate estimate is needed.

To carry out numerical diffusion diagnostics presented above, we divide the available salinity range ([0, 30]) into 200 classes

with the isohaline values $S_i = S_r + i\delta$, $i = 1 \ldots 200$, where $S_r = 0$ is a salinity of the river water and $\delta = 0.15$ is a size of the salinity class.

**Figure 9.** The salinity-volume diagram for runs 1, 2 and 7 at different time moments. The dashed and solid lines indicate the solutions at 20h and 35h correspondingly.

The salinity-volume diagram in Figure 9 allows us to trace the total volume of each salinity class and in particular the volume of the first freshwater class. Generally, numerical diffusion causes the appearance of volumes between the first and last salinity classes (contain fresh and shelf saline water masses respectively), by reducing the volume of the first salinity class and, in our case, of also the last salinity class (no open boundaries are presented). If the total volume of the freshwater class is less than the volume of the river channel ($5 \cdot 10^7$ m$^3$) it means that the dense water penetrates into the river channel (Fig. 9). Note that with zero eddy diffusivity, all intervening classes (except for the classes, which can be attributed to the layer occupied by the plume) have approximately the same volume compared to each other for the bulk of the solutions (shown only for runs **1** and **2**), indicating that numerical diffusion works through salinity classes. (Our statements about volume should be interpreted taking into account that the *total* volume of each salinity class is considered.)



**Figure 10.** The mean (over second inertial period) area of the surface projection of the isohalines for different runs.

The significant shift in the diagram can be attributed to the choice of the advection scheme; a low-order advection scheme tends to make a layer, occupied by the plume, saltier (Fig. 9): in run **2** the plume initiated surface layer is a bit saltier compared to run **1** and has a different structure. When eddy diffusivity is on (run **7**), the volume of the higher salinity classes increases with time while remaining stable and very small across the salinity classes characterised by salinity less than 10. It takes place because in this case we have a pronounced mixed layer with a salinity of about 10 in the near-field, bulge and coastal current zones (Fig. 5). Also note that despite the qualitative similarity of diagrams for runs **1, 11** and **12** (not shown), the total volumes of the first salinity class are more than on $34\%$ at 20h and 35h for the runs performed on a rectangular grid (runs **11, 12**).





Runs **11** and **12** provide maximum freshwater volume for the first salinity class among all runs. In case of runs **1, 11** it is signalizing about the fact, that the quasi-B discretization on the triangular grid can be a noticeable source for numerical mixing
unless noise is suppressed by a filter. In case of run **12** we can say that anti-diffusion/anti-viscous *superbee* limiter (for tracer and momentum advection in GETM) is highly effective in the vertical in conserving the two-layer system with pronounced interface between plume and quiescent layer (Fig. 8c, d).

It is known (e.g., Hetland, 2005; Burchard, 2020) that a larger isohaline area means less mixing for a given level of freshwater transport; so the total volume of a particular class may be the same while the mixing level differs from case to case. The volume
of the first class may also be larger from one case to another in line with a higher numerical diffusivity level; for example, this would apply where a plume spreads relatively little and remains relatively thick. To avoid wrong interpretation of the salinity-volume diagram, the area of the isohalines (Fig. 10) for the different runs should be considered.

The shape of all curves except for run **7**, which is characterized by the presence of physical eddy diffusivity, are very similar. The shape reflects that a two-layer system is considered. The curves, which are characterized by a larger area of the isohalines,
mean larger offshore spreading of the plume. Even so the coastal current in the more diffusive solutions can propagate faster, it is less restricted offshore together with a bulge (e.g., Fig. 3). The shape of the curve for run **7** with eddy diffusivity turned on signalizes about the presence of the homogeneous, in a sense of salinity, layer occupied by the plume, which is thicker compared to the same run without physical eddy diffusivity (run **1**). It can already be seen that the limiters, which are strictly preserved the monotonicity of the advection scheme, effectively reduce its order: the curves for runs **4** and **5** are nearly the
same (Table 1). The vertical resolution also plays a large role, logically the coarse vertical resolution introduces more numerical diffusion (see below), so as a result the curve for run **10** is significantly below the curve for run **5**, and the layer occupied by the plume is saltier in run **10** compared to run **5**. Also Figure 10 indicates the salinity, which can be used as a representative for the layer occupied by the plume (as an example it can be the first inflection point; hence, for run **7**, it is the salinity of ∼10). The curve for run **3** crosses the curve for run **1**, which means that the plume layer is not so fresh in run **3** as in run **1**, but the
plume spreads a bit further offshore, what we can already seen in Figures 3, 5.



**Figure 11.** Analysis of numerical diffusion in the system: a) Total freshwater discharge trough different isohalines, $\overline{F}$, m$^3$/s; b) Transport per unit area of each isohaline, $\overline{f}$, m/s; c, d) Total salinity fluxes through different isohalines, $\overline{f^s}$, psu m/s; the numbers indicate the sum of the salinity fluxes multiplied by $10^3$. e,f) Diahaline velocities, $\overline{\frac{\partial f^s}{\partial S_i}}$. In all pictures the characteristics are averaged over second inertial period.

Despite the fact that the Figure 9 and Figure 10 indicate that the diffusive processes take place, there are several aspects to be clarified. For example, we claimed that run **12** has one of the largest freshwater volume, however, in Figure 10 the curve for



run **12** starts below than for runs **1, 3** and **4**. Also quantitative estimation of the level of numerical mixing requires additional steps to be made. Therefore we make use of the characteristics introduced in the Numerical Diffusion section.

We will start consideration from the total salinity discharge trough different isohalines divided by the salinity of the current isohaline (named as 'freshwater discharge', Fig. 11a). We should note that freshwater discharge largely demonstrates the performance of the runs in vertical. The total discharge through relatively fresh isohalines (Fig. 11a) caused by numerical diffusive processes contains a minimum at ∼-3000 m³/sec, which means that nearly the total river discharge is transported through this isohaline, increasing the volume of higher salinity classes. The non-monotonic solution (no limiters are introduced

for the tracer advection scheme) can be readily seen: run **6** accepts the discharge larger then ∼-3000 m³/sec. For the less diffusive solutions the absolute value of the discharge than rapidly decreases, signalizing that the relatively fresh surface layer is forming (the isohaline layer is growing and accumulating the freshwater). For run **7** the discharge is nearly equal to ∼-3000 m³/sec until the salinity class of ∼10, so the plume forms the layer with a salinity of ∼10 even at the surface. The curve for run **13** has similar shape. It signalises about the fact that the limiter used for vertical advection works quite similarly to a physical

vertical eddy diffusivity. The smallest numerical freshwater discharge can be attributed to the runs performed on rectangular grid (runs **11, 12**), with a best result attributed to the run **12**. It means that the layer occupied by plume is fresher compare to others solution. This explains large freshwater volume in the system with a good but relatively moderate offshore spread of the plume, which was described above. Also the noise originated due to the type of discretization in case of run **1** compared to run **11** slightly damps the dynamics in the vertical but not in horizontal direction (Fig. 10, Fig. 11a). The discharge for the

run with eddy diffusivity turned on is more efficient in mixing the fresh water into the higher salinity classes and increasing their volumes (Fig. 9). This can be seen from the lowest position of the curve of run **7** in Fig. 11a. However, there are several classes of higher salinity, where diffusion is small compared to all other runs, because nearly all discharge has been already 'dissolved' at lower salinity classes. Interestingly, the curve for run **2** lies somewhere between the curves for runs **5** and **10**. It means that for this particular task more vertical layers should be introduced in case of a DG model.

Figure 11b shows that the transport is tending to decrease when the considered isohaline increases, even so the discharge can be the same (e.g., run **7**). This follows from the fact that the isohalines of higher salinities have larger areas. Figure 11b shows that run **2** has a large cross-isohaline transport for low salinities, therefore despite its generally good performance it is characterised by a blurry surface layer (Fig. 3). Also we see the difference between runs **7** and **13**: the transport trough the freshwater isohaline is largest for **13** but then rapidly decreases - there is no pronounced mixed layer.

As we already mentioned, Figure 11a can lead to misunderstanding because the *total* freshwater discharge is considered. To have a transparent diagnostic we draw the total salinity discharge trough different isohalines normalized by the isohaline area (Fig. 11c, d). In other words Figures 11c, d show the averaged diffusive salinity fluxes per unit area of the particular isohaline (averaged numerical diahaline transport per unit area of each isohaline), which makes it easy to identify more diffusive numerical solutions. With eddy diffusivity turned off we have zero physical diffusive transport, so the corresponding curves

demonstrate purely numerical diffusive transport, the work of the advection scheme and limiters, time stepping and diffusive filters. With non-zero eddy diffusivity, the diahaline salinity transport curve contains physical and numerical components. Note that activating eddy diffusivity contributes to a smoother velocity field, so an 'absolute' numerical diffusivity (with and without





eddy diffusivity) is out of question. But turning on eddy diffusivity in this particular test-case will generally lower the numerical diffusivity (see the second local minimum of the solution); physical eddy diffusivity reduces the numerical diffusivity of the advection scheme. The numbers in Figure 11c show the sum of the salinity fluxes multiplied by $10^3$, so they demonstrate the average level of numerical mixing for the current solution. Run **2** in comparison with runs **1, 4** and **5** demonstrates the larger diffusion for the layer occupied by the plume and smaller level of diffusion for the higher salinity classes and even anti-diffusion for the last salinity class. We speculate that the vertical resolution within the layer occupied by the plume is insufficient. A similar conclusion was already made when we inspected the surface salinity in Fig. 3. Run **10** clearly demonstrates that the coarser resolution causes a higher numerical mixing level and a blurry top plume related layer. Here, we would like to stress that runs **11** and **12** have the smallest level of numerical mixing. Here, we exclude run **3** with *fct2* version of the limiter from considerations, in run **3** the level of numerical mixing is very close to run **1** with *fct1* option, but slightly smaller (the sum of salinity fluxes is equal to -4.9 · $10^{-3}$ psu m/s). Note, the *fct2* option works much worse in case of rectangular grid, reacting to a relatively coarse grid (not shown, the sum of salinity fluxes is equal to -7.2 · $10^{-3}$ psu m/s).

Some interesting and not obvious behaviour of the system can be tracked by looking at the velocities through different salinity classes (Fig. 11d, e). It can be noticed that more diffusive solutions have a positive peak at low salinity classes. It means that the current class is largely growing in a sense of volume due to the presence of the discharge. Some runs, in particular, **1, 4, 5, 11, 12** do not have this feature, demonstrating nearly symmetric transport through salinity classes.

All models (e.g., runs **1, 2** and **12**) track the total freshwater volume in the system accurately. Run **2** tends to underestimate the volume by roughly 2%. The discrepancy is due to the weakly imposed river boundary condition in Thetis; the model itself is mass conservative. The applied advection schemes with limiters (except run **6**, which is not equipped with limiter; Table 1) are nearly monotone, the over- and under- shoots for runs **1, 2, 3, 5, 7-11** for salinity after 20h of simulations are not more than $10^{-10}$ in practical scale; runs **4, 12, 13** are strictly monotone.

## 6  Discussion

In this section we discuss the sensitivity experiments and interpretation of the results. In particular, the effect of vertical resolution, penetration of the dense water into the river channel and the role of vertical viscosity on the simulated plume behaviour are clarified.

### 6.1  Vertical resolution

Runs **9** and **10** with a reduced number of sigma layers (20 instead of 40) show reduced bulge offshore spreading by 7-8 % (Fig. 12). Also, even though runs **9** and **10** use different advection schemes (the hybrid MUSCL-type and Miura, respectively), they simulate a more saline surface layer compared to the runs **1** and **5**. In case of run **9** the bulge is less restricted offshore and thicker at the same time - the coastal current is reduced together with offshore spread of the bulge - compared to run **1**, but there is no pronounced redistribution of the discharge rates between bulge and coastal current. Numerical mixing increases by 26 % in run **9** compared to run **1** (the sum of salinity fluxes of numerical origin is equal to -6.3· $10^{-3}$ psu m/s for run **9**). In





case of run **10** there is also a bit thicker plume and also reduced discharge rate goes to the bulge and an increased discharge rate

goes to the coastal current. Compared to run **5** the bulge turns out to be largely sliced off by the coastal wall and the coastal

current propagates further. The numerical mixing increases by 35 % in run **10** compared to run **5** (the sum of salinity fluxes is

equal to -7.8·$10^{-3}$ psu m/s for run **10**).

    As expected, a coarser vertical mesh increases numerical mixing which in turn modifies the plume behaviour and final

characteristics. It also restricts the minimum thickness of the plume. However, this effect is minor due to the zooming towards

the surface of sigma layers and the limited water depth (it does not exceed 30 m).

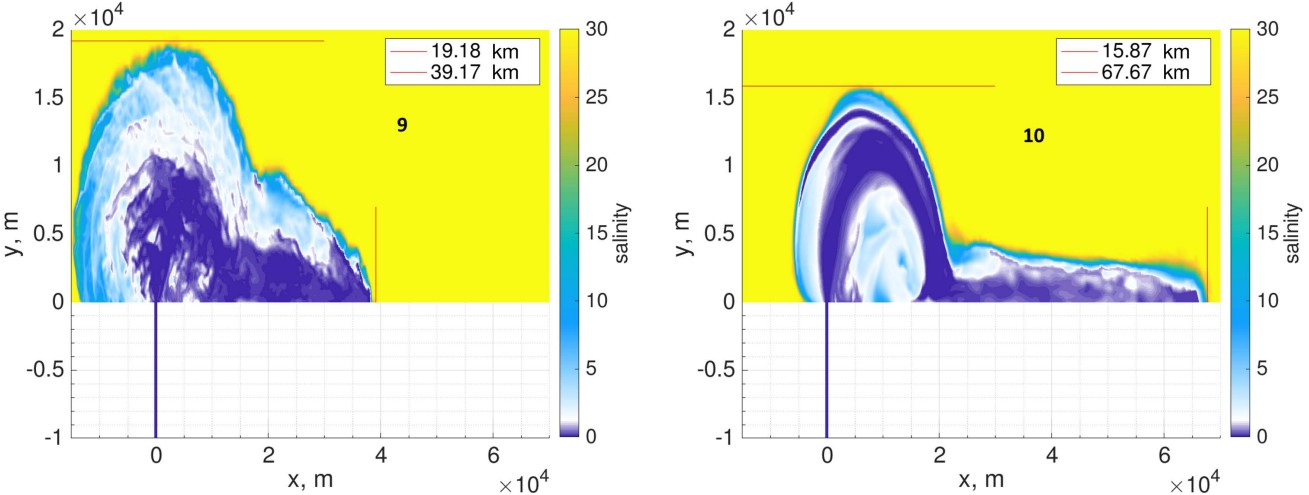

**Figure 12.** The surface salinities at 35h, in practical scale, as the results of the simulations **9** and **10** with reduced amount of sigma layers
compared to the simulations **1** and **5**.

## 6.2 Penetration of the dense water into the river channel

The dense water penetration into the river channel is sensitive to the detail of simulations. Since the resolution of triangular

and rectangular grids is similar in the vicinity of river mouth (Fig. 2), we concentrate on other factors. We have shown that

numerical mixing related to limiters or upwind fluxes in the advection scheme can cause the penetration of the dense water into

the river channel. However, run **1** has a larger level of numerical mixing compared to run **12** but does not contain dense water

in the river channel opposite to run **12**. Therefore, the relationship between numerical mixing and penetration depends on the

source of numerical mixing. As we already mentioned the increased level of numerical mixing in run **1** compared to runs **11,**

**12** is likely related to residual effects of spurious inertial oscillations supported by the discretization type.

The area of hydraulic jump and shelf-river channel interface are also highly sensitive to the discretization of momentum

equation, in particular to calculation of the pressure gradient, momentum advection and vertical viscosity. The latter is of prime

importance. According to hydraulic theory the penetration of the dense water should occur if the liquid is inviscid. Obviously

the vertical viscosity works oppositely to numerical mixing in this respect: relatively high viscosity blocks penetration of the





dense water into the river channel. We made several sensitivity runs when only constant vertical background viscosity was
present, and no limiting was applied in the vertical advection of momentum. Figure 13 visualizes some of them (runs **8a,b,c**;
Table 1). This demonstrates the dynamics of the plume with a standard set of options on a triangular grid (as in run **1**) except for
the vertical viscosity (the horizontal viscosity was set to zero). If the viscosity equals $3 \cdot 10^{-4}$ m$^2$/s, the penetration takes place
and has a typical lock-exchange profile. When the viscosity is increased to $10^{-3}$ m$^2$/s, a much slower penetration is simulated.
For an extreme value of background vertical viscosity of 0.1 m$^2$/s, no penetration is seen.

Among all runs only run **12** has the limiting for the advection of momentum. The superbee limiter was used, characterised by
locally anti-viscous (anti-diffusive) behaviour, which can affect the simulated penetration of dense water. As we have already
mentioned, numerical mixing can naturally provoke the dense water penetration. Note, that in all cases with relatively lower-
order (up to second order spatially) advection scheme the penetration occurs and has a blunt face profile (runs **2, 5, 6, 12**; Fig.
3, 5). We can conclude that in run **12** the penetration occurs due to usage of the advection scheme of a second order for tracer
and anti-viscous limiter for the advection of momentum.

We repeated the runs, which showed the penetration of a dense water into the channel, with the discharge increased to
3900 m$^3$/s. As expected (see section with analytical prediction) all runs with new discharge do not have penetration of dense
water into the river channel. Also we should note that in the absence of bottom friction the penetration did not influence the
plume characteristics considered by us. For example, run **12** has slightly larger offshore spread - about 1 km - compared to old
realization, as it should be according to the analytical solution (Fig. 14).



**Figure 13.** The surface salinity and salinity at the mouth transect at 35h with different background viscosity levels (runs **8 a,b,c** from top to bottom).

## 6.3 Viscous processes

This subsection is largely based on an additional set of experiments with a slightly increased discharge of 3900 m$^3$/s and a constant vertical viscosity equal to $2.5 \cdot 10^{-4}$ m$^2$/s (Table A1; for convenience, the old numbering is preserved). The slightly larger



discharge is preferable because it results in thicker plume layer. The analytical prediction for the major plume characteristics

in this case is given in brackets in the corresponding section.

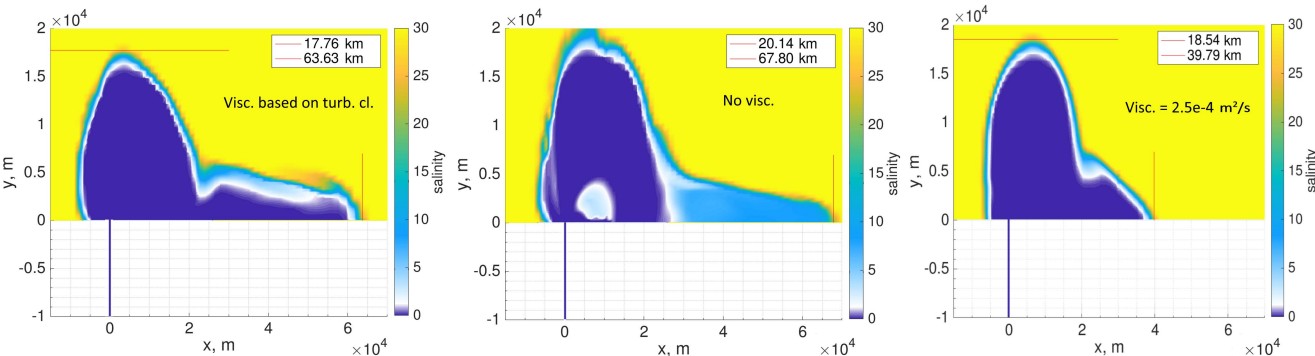

**Figure 14.** The surface salinity, in practical scale, at 35h as a result of the runs **12c, 12b, 12** ( see Table A1; discharge rate is 3900 m$^3$/sec).

The experiments described above make clear that the vertical viscosity and the details of the momentum equation discretization play a significant role in the simulated plume dynamics. The analytical solution approximates the flow as a two layer system with a plume layer having an uniform shear and a motionless lower layer. If the vertical background viscosity in simulations is lower than 2e-5 m$^2$/s (the horizontal viscosity is set to zero), all models participating in this study cannot preserve

the two-layer system, the coastal current has an average salinity above 10 and anomaly large speed. An example is given for run 12, Fig. 14, middle panel, which gives one of the best results, but still contains these artifacts. It could also be that stability conditions cannot be met for a reasonable time step above 1 s for the baroclinic mode. The reason is the presence of numerical mixing in the system, which dilutes the thin freshwater layer, and other numerical inaccuracies, which have in this case pronounced consequences. With the vertical viscosity of about $10^{-3}$ m$^2$/s the solutions are significantly damped: the

offshore spread of the plume is restricted together with reduced coastal current, the plume layer is too thick (e.g., Fig. 13). If the viscosity has an order of $10^{-4}$ m$^2$/s, all solutions are relatively close to each other, compared to the runs with turned on eddy viscosity, as concerns the offshore plume spreading and the nose position in the coastal current (Fig. A2). Note that with this level of background viscosity runs **1** and **11** are much closer to each other as concerns the level of numerical mixing and offshore spread of the plume. This means that the noise triggered by triangular grid in FESOM-C is already partly damped.

The eddy viscosity calculated based on turbulence closures (see, e.g., runs **1b, 12c**; Table A1) in the area of plume spreading has mean value of an order of $10^{-4}$ m$^2$/s, with the maximum reaching an order of 10 m$^2$/s. This naturally means that the background value of an order of $10^{-4}$ m$^2$/s is too small and too large for the different subareas compared to the eddy viscosity based on turbulence closures (see, e.g., Fig. A1).

For the runs with only background viscosity the level of numerical mixing and agreement with the analytical solution are

not in a direct correspondence: in this case the differences between the runs are largely reflected in the volume of intermediate classes, which are to a degree separated from the surface dynamics if there is no eddy viscosity. So, the new run **6** has fresher surface and larger offshore spread compared to run **2**, but a higher level of numerical mixing (see, e.g., Fig. A3, A4). Also, for





example, run **9** with a twice smaller number of vertical levels shows better performance with respect to the level of numerical mixing and offshore spread compared to run **1** with the same set of options, but with the larger number of layers. Run **12** with
anti-viscous and anti-diffusive limiters shows the best performance among all runs in the sense of numerical mixing (Fig. A4).

We note that the thickness of the plume layer at the coastal current has a different structure in all these runs compared to the runs with eddy viscosity: in particular, it does not reach a maximum near the wall as in analytical and laboratory studies (see Fig. A2). An additional issue is the absence of the bulge symmetry predicted by the laboratory studies (they were conducted with viscid liquids), in particular, too large offshore extent of the bulge compared to its alongshore size. All numerical solutions
have different built-in viscosity and it is hardly possible to find one value of background viscosity which would account for the difference without damping the dynamics significantly. The eddy viscosity based on turbulence closure works toward zero stress in the plume layer (large stress in surface layers is generated by a large pressure gradient). When the eddy viscosity is turned on, the details of momentum-equation discretization become less important (as in the case of eddy diffusivity, the presence of physical eddy viscosity makes the level of numerical viscosity smaller) and the level of numerical mixing can
be estimated by offshore plume position. The last point was crucial for the current test case, which has largely a focus on numerical diffusivity. We therefore suggest to use turbulence closure for eddy viscosity instead of constant viscosity in this test case.

Note that a decrease in momentum rather than the increase in density during plume entrainment is responsible for the decrease in Froude numbers (Hetland, 2005). Entrainment acts to force the flow toward criticality (Gerdes et al., 2002). Therefore,
for the absent eddy diffusivity we mostly retain the salinity gradient but the Froude numbers nevertheless decrease upstream from the supercritical area. So, the choice of turbulent closure for the vertical eddy viscosity coefficient impacts the way different zones are functioning. For this reason we highly recommend employing second-order turbulence models (e.g., see review in Umlauf and Burchard (2005)) to be able to compare carefully results with the results presented here.

We mention that the presence of the supercritical conditions in addition to the numerical mixing can be a reason for the
underestimations of the bulge offshore spreading by the numerical solutions. However, the analytical solution is given for the wide time range (one inertial period), this ensures the feasibility of the numerical solution to reach analytical one. The prime results of runs **11** and **12** demonstrate that (Fig. 8).

### 6.4 Model type

In the test case three models with different discretizations have been applied: run **2** was performed using the Thetis DG finite
element model, that uses formally a second-order advection scheme and geometric slope limiters; other runs except runs **12, 13** were performed based on the FESOM-C, B-grid finite volume model with different advection schemes and flux limiters. Runs **12, 13** were performed based on the GETM C-grid model. Primarily, the dynamics is defined by the order of the advection scheme and the quality of the limiters and not by the model type. However, some differences in the dynamics can be found due to the model and discretization type. In case of the DG model the gradient of the tracer field is noisier compared to the velocity
field, in case of the finite volume model with quasi-B discretization the velocity field contains noise on the triangular grid. The problem is well known, the solution is given by applying biharmonic operator, see, e.g., Danilov and Androsov (2015).





Therefore, the thickness of the plume layer was identified differently (based on velocity and salinity fields in case of DG and finite volume solution respectively), however, in the end the border of the plume layer was going through the same isoholine in both models. Also, the DG model is less diffusive in the layer not occupied by the plume and more diffusive in the plume layer

compared to the GETM and FESOM-C runs with the second order (horizontally) advection scheme. As a result in the Thetis run the surface is saltier and smoother, but the system has pronounced two-layer character and the layer occupied by plume is not thick. The level of numerical mixing and general the behaviour of the salinity in the plume layer in run **2** is comparable to the reduction of the amount of vertical layers in run **5** (run **10**). There is no obvious answer on this issue, most probably, it is a combination of the vertical discretization, current advection scheme and limiters performance within the particular task.

## 755 7 User information

### 7.1 Summary of the test case, major points

The test case presents a river-shelf system: there is a source of the freshwater and momentum separated from the shelf by a perpendicular river channel, there are no additional forcing terms and no open boundaries (whole volume, which is coming into the system, stays there). In the numerical solution the eddy diffusivity should be put to zero, which makes the diagnostic of

the numerical mixing very transparent and allows us to consider the plume behaviour at some aspects in frame of a two-layer paradigm (see analytical prediction section for the details). The eddy viscosity coefficient should work and be calculated based on second-order turbulence models (e.g., see review in Umlauf and Burchard (2005)), preferable of $k - \epsilon$ style. The test case assumes the usage of linear equation of state. However, if the user does not have assess to the equation of state and would like to try non-monotonic advection scheme or advection schemes with flexible limiters, we suggest, for example, to use the

salinity 2 and 32 for the river and shelf waters respectively to avoid negative salinity in the numerical solutions at the frontiers areas, the most important is to keep the density difference between the river and shelf water masses, which should be equal to 23.01 kg/m$^3$. In this case the pictures should be the same as presented in the current article, only the colorbar or salinity range should be shifted toward high salinity. The bottom friction should be turned off. In the simulation 40 sigma layers (41 levels) should be prescribed with the parabolic distribution (zooming to the surface). If for some reasons the last point can not

be reached then the running of test case still makes sense, however the numerical diffusivity level given in the article should not be used as a reference.

The total simulation time is 35 h, the grid contains $\sim 76 \cdot 10^3$ triangles ($\sim 60 \cdot 10^3$ rectangles), so the task can be calculated within appropriate time even by models running on laptops.

The further details of the setup are specified in the Modelling Setup Section.





## 7.2 Summary of the analytical solution

After $\sim$ 8.2 h of the total simulation time the plume is expected to start turning right at an approximate distance of 5 km from the coast. After 20-35 h the offshore spread of the bulge should reach $\sim$ 24 km. The surface layer occupied by plume should be fresh.

The estimated maximum and minimum thicknesses of the coastal current are about 1.8 m and 0.9 m respectively, and maximum velocity (the near-wall speed in the layer occupied by the plume) is in the range from 0.45 to 0.64 m/s. The cross-shore width of the coastal current can vary from 3.75 to 12 km after one to two rotation periods.

The surface salinity, profile of dense water intrusion into the river channel (of presents) and its speed, coastal current's discharge rate, and offshore spread as well as asymmetry or the characteristic impingement angle of the bulge can all be considered as indirect measures of numerical mixing. However, simplest way is to look on the offshore spread of the bulge and its surface salinity. Any source of numerical mixing reduces offshore spread of the bulge or/and makes its surface saltier. Numerical mixing pushes bulge to be less restricted offshore and in parallel deeper or/and the bulge tends to be less restricted offshore together with reduced discharge rate goes to the bulge and increased discharge rate goes to the coastal current (bulge turns out to be sliced off by the coastal wall).

## 7.3 Output requirements and corresponding definitions

In the current test case we are focusing on the final characteristics of the plume spreading, in particular the position of the lift-off zone, bulge characteristics at a given time (offshore, nearshore-wide and alongshore-long), depth of coastal currents, its cross-front width, and velocity. These characteristics should be extracted at two time moments: 20h and 35h from the beginning of the simulation time (20h is approximately the sum of one inertial period with time period, which river needs to get to the mouth area (1)).

For users, who are interested in detailed analysis of the model performance we recommend to fill the Table 2 (the empty table - only with analytical prediction - is provided in Supplementary) and to perform full numerical diffusivity analysis or at least get the number provided at Figure 11c.

Tracing the level of numerical diffusive transport and checking the prediction against the analytical solution requires next output: a) surface velocities and elevation and 3D salinity fields for two moments in time—20 h and 35 h from the beginning of the simulation period (the time it takes for the river to reach the mouth area having already been accounted for); and, b) velocity profiles for two transects at the specified times (Fig. 2). The calculation of the mean isopycnal areas within second inertial period can be simplified to the calculation of the surface projection of each considered isohaline (sum of vertex or, depending on the type of discretization, element or edge control areas, where particular salinity S is present at any depth) at two time moments - 20 h and 35 h and its further averaging. The full range of salinity values should be splitted into 200 salinity classes with the salinity difference between neighboring classes equaled to 0.15. The first and last classes can be larger if non-monotonic advection solution is used (it is important to cover whole salinity range presented in the model).





We should also specify here how the bulge and coastal current parameters should be calculated: the layer-thickness occupied by the plume is defined by the maximum salinity or velocity gradient depending on the discretization (one field can be noisier than another); the shape of the bulge is identified by the surface plot of salinity: the salinity less of 29 in practical scale should be considered as a border to identify the bulge spread and coastal current position.

The shortest version of the analysis includes only surface visualization of the salinity at 35 h time moment. Ideally in the system after 35h the bulge offshore restriction should be more than 24km (and coastal current nose should not reach 40 km on the $x$-scale), the layer occupied by a plume should be fresh. Minimum acceptable result is that the bulge spreads offshore at about 16 km (and coastal current does not not reach 70 km).

Additionally the total water volume in the system should be checked after 20h (some limiters can artificially reduce the river inflow to stabilize the system). It should increase by 0.2052 km$^3$ plus amount of the freshwater, which flows during 1st hour - 0.0054 km$^3$.

## 8 Code and data availability

GETM is an open-source coastal ocean model (Burchard and Bolding, 2002, www.getm.eu), the code and installation instructions are available from https://getm.eu/software.html (last access: 23 December 2020). The version of FESOM-C v.2 (Androsov et al., 2019) used to carry out the simulations reported here can be accessed from https://doi.org/10.5281/zenodo.2085177 (last access: 23 December 2020). Thetis code (Kärnä et al., 2018) used to perform the experiments is also publicly available. It may be obtained from http://thetisproject.org/ (last access: 23 December 2020).

The experiments data can be downloaded from http://doi.org/10.5281/zenodo.4389353.

The grid files and table to fill are attached to the article as Supplementary materials.





**Appendix A**

| Number | Adv. scheme | Turbulence closure for tracer equation | Limiter | Viscosity, m²/s | N of vertical sigma levels with parabolic distr. | Model/grid |
|---|---|---|---|---|---|---|
| 1 | 85% of 3rd order + 15% of 4th order | | fct1 | 2.5 e-4 | 41 | FESOM-C/tri |
| 1b | 85% of 3rd order + 15% of 4th order | | fct1 | turb. closure | 41 | FESOM-C/tri |
| 2 | 2nd order (upwind) | | geom. lim | | 41 | Thetis/tri |
| 4 | 85% of 3rd order + 15% of 4th order | | fct3 | | 41 | FESOM-C/tri |
| 5 | 2nd order (Miura) | | fct1 | | 41 | FESOM-C/tri |
| 6 | 2nd order (upwind) | | no | | 41 | FESOM-C/tri |
| 9 | 85% of 3rd order + 15% of 4th order | off | fct1 | 2.5 e-4 | 21 | FESOM-C/tri |
| 10 | 2nd order (Miura) | | fct1 | | 21 | FESOM-C/tri |
| 11 | 85% of 3rd order + 15% of 4th order | | fct1 | | 41 | FESOM-C/quad |
| 12 | 2nd-order (TVD) | | superbee | | 41 | GETM/quad |
| 12b | 2nd-order (TVD) | | superbee | inviscid | 41 | GETM/quad |
| 12c | 2nd-order (TVD) | | superbee | turb. closure | 41 | GETM/quad |
| 13 | 3d order HSIMT (TVD) | | Sweby's lim. | 2.5 e-4 | 41 | GETM/quad |

**Table A1.** The description of the additional runs, the discharge is equal to 3900 m³/sec (other setup details are the same).



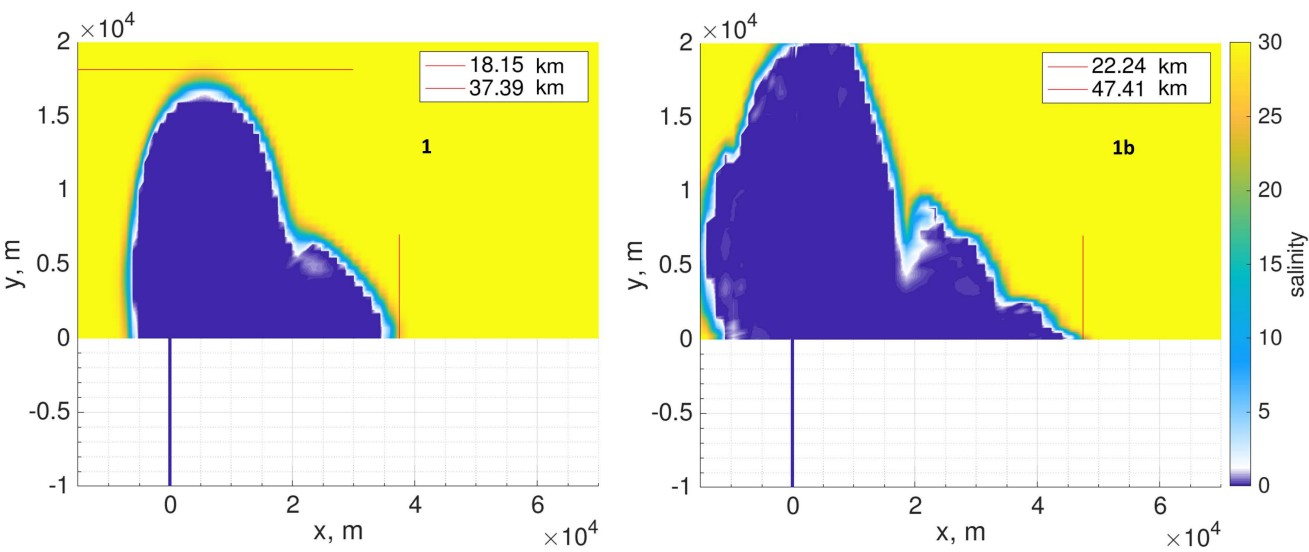

**Figure A1.** The surface salinity, in practical scale, at 35h as a result of runs 1 (left panel) and 1b (right panel).


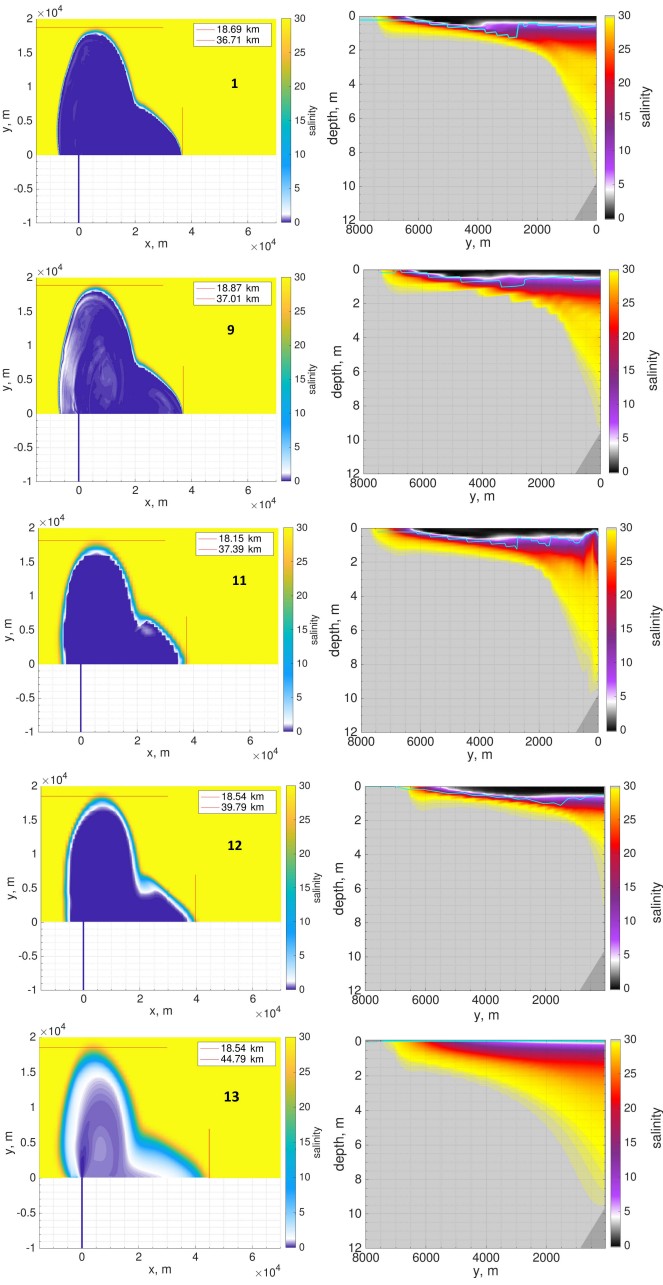

**Figure A2.** Salinity, in practical scale, as a result of the different runs at 35h at the surface (left panel) and coastal current transect (right panel).





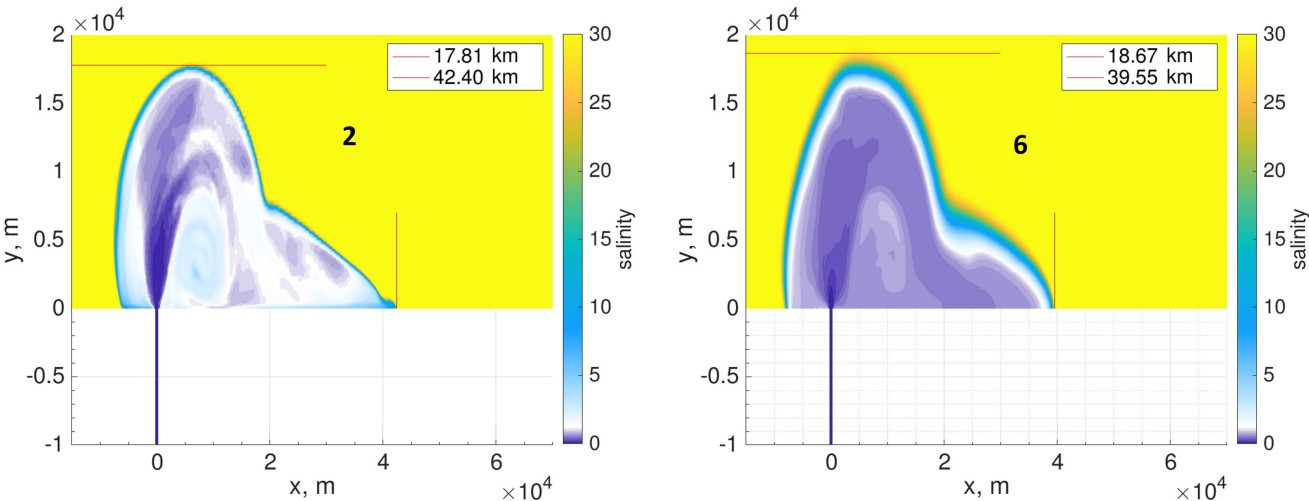

**Figure A3.** The surface salinity, in practical scale, at 35h as a result of runs 2 (left panel) and 6 (right panel).

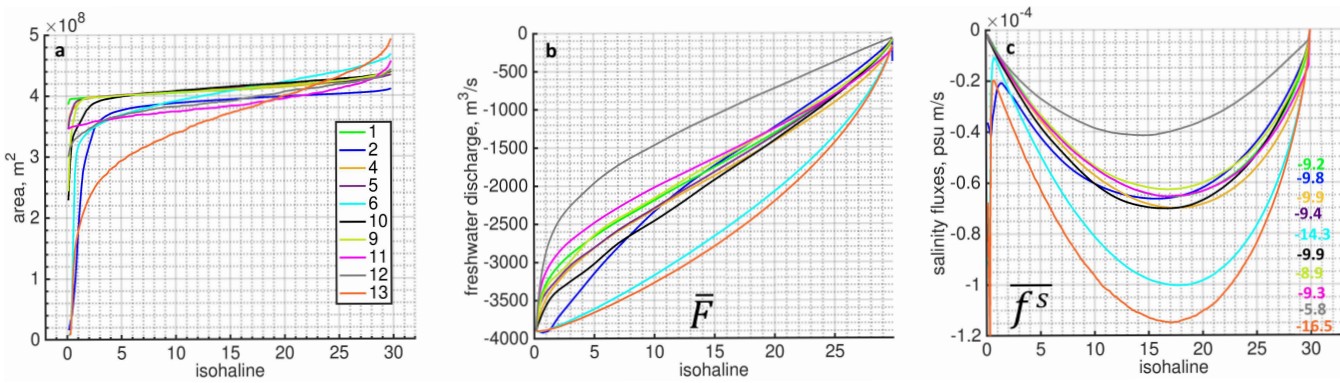

**Figure A4.** Analysis of numerical mixing in the system performed for the different runs (Table A1): a) Isohaline areas; b) Total freshwater discharge trough different isohalines, $\overline{F}$, m$^3$/s; c) Total salinity fluxes through different isohalines, $\overline{f^s}$, psu m/s; the numbers indicate the sum of the salinity fluxes multiplied by 10$^3$. In all pictures the characteristics are averaged over second inertial period.

*Author contributions.*  VF built a test case, its analytical solution, description, structure and general framework of the analysis and prepared the results. TK actively participated in improvement of the test cases in all aspects from set up to analysis and of the text, created final triangular grid, made run 2, took part in all discussion rounds, wrote the Thetis model description, prepared Figure 1 and output from run

830   2. KK made valuable and extensive comments through the whole manuscript, prepared several runs and its output and GETM description, initiated activity on the current test case. AA prepared rectangular grid and FESOM-C model description, actively took part in discussion. SD and HB controlled scientific quality of the paper, improved significantly the structure of the paper and took part in the results discussion.



IK and DS provided very valuable technical support, improved significantly the text of the manuscript and gave valuable comments about structure of the paper. KHW supervised the work.

835 *Competing interests.* Authors declare that no competing interests are present to their knowledge.

*Acknowledgements.* We would like to thank Natalja Rakowsky and Sven Harig for the valuable technical support. Also we are grateful to Peter Arlinghaus for the valuable comments and remarks at the beginning of current work.



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
