# Peer review of "Plume spreading test case for coastal ocean models"

_Geoscientific Model Development, 2020_

## Author Comment (AC4)

**MEETING SUMMARIES**

**ADVANCING DYNAMICAL CORES OF OCEANIC MODELS ACROSS ALL SCALES**

Florian Lemarié, Hans Burchard, Laurent Debreu, Knut Klingbeil, and Jacques Sainte-Marie

Oceanic numerical models are used to understand and predict a wide range of processes from global paleoclimate scales to short-term prediction in estuaries and shallow coastal areas. One of the overarching challenges, and the main topic of the Community for the Numerical Modeling of the Global, Regional, and Coastal Ocean (COMMODORE) workshop is the appropriate design of the dynamical cores given the wide variety of scales of interest and their interactions with atmosphere, sea ice, biogeochemistry, and even societal processes. The construction of a dynamical core is a very long effort that takes years and decades of research and development, and requires a collaborative mixture of scientific disciplines. This work involves a significant number of fundamental choices, such as which equations to solve, which horizontal and vertical grid arrangement is adequate, which discrete algorithms allow jointly computational efficiency and sufficient accuracy, etc. Nowadays, a

**AFFILIATIONS:** Lemarié and Debreu—Université Grenoble Alpes, Inria, CNRS, Grenoble INP, LJK, Grenoble, France; Burchard and Klingbeil—Leibniz Institute for Baltic Sea Research Warnemünde, Rostock, Germany; Sainte-Marie—Inria Paris, and Sorbonne Université, Laboratoire Jacques-Louis Lions, Paris, France

**CORRESPONDING AUTHOR:** Florian Lemarié, florian.lemarie@inria.fr

DOI:10.1175/BAMS-D-18-0303.1

In final form 15 November 2018

©2019 American Meteorological Society
For information regarding reuse of this content and general copyright information, consult the AMS Copyright Policy.

**FIRST COMMODORE WORKSHOP: COMMUNITY FOR THE NUMERICAL MODELING OF THE GLOBAL, REGIONAL, AND COASTAL OCEAN**

WHAT: A total of 47 participants from 9 countries representing 15 different oceanic numerical models met to review our current understanding of future challenges in the design of oceanic dynamical cores.

WHEN: 17–19 September 2018

WHERE: Paris, France

broad range of numerical methods are implemented in models used for realistic ocean simulations, and, owed to the advances in computational power, a meeting point has been reached between global circulation models and regional local models, such that there can be mutual benefits of a cross-fertilization between communities. This report outlines an initiative to bring together the worldwide leading researchers actively contributing to the development of oceanic model dynamical cores, such that participants could network together and focus on next challenges irrespective of target applications (regional, coastal, or global). The first COMMODORE workshop (https://commodore2018.sciencesconf.org/) was organized in Paris, France, in September 2018. In total, the participants represented 15 oceanic dynamical cores among the most widely used by the research and operational community. The motivations, topics of discussion sessions, and outcomes of the workshop are summarized below.

**CONTEXT.** The ocean model developers community has had the tendency to be split depending on the target applications (global vs coastal) and on the type of horizontal grids (structured vs unstructured) and has been organized around relatively small modeling groups. However, the models have now reached such a high level of complexity that model development goes beyond the expertise of one given group and requires interactions between physicists, mathematicians, and computer scientists. In this context, this workshop aimed at gathering a community of "model oriented" researchers to foster more regular exchanges and share expertise on outstanding issues and perspectives. During this first workshop, the emphasis was on reviewing the characteristics and diversity in the formulation of oceanic models used for realistic applications as well as on outlining upcoming challenges.

**EVOLUTION OF OCEANIC MODELS.** Historically, global and regional ocean models have been based on the hydrostatic primitive equations (e.g., Griffies and Adcroft 2008) discretized on structured grids using a mixture of finite-difference and finite-volume techniques for the discretization in space. The time dimension is usually treated using standard predictor–corrector or two-level approaches (e.g., Lemarié et al. 2015b). Those choices have been made because of their good compromise between simplicity, efficiency, and accuracy. In recent years, significant progress has been made for ocean modeling on unstructured grids, either via the finite volume (e.g., Chen et al. 2003; Ringler et al. 2010; Danilov et al. 2017) or finite element (e.g., Zhang et al. 2016; Korn 2017; Kärnä et al. 2018) approach. Unstructured grid models have reached an unprecedented level of maturity at least for two reasons. First, the vertical dimension is treated in a structured way compared to earlier initiatives trying to get three-dimensional unstructured meshes working. Second, a better understanding of computational modes and dispersion properties associated with a wide range of possible choices of finite-element pairs has been reached (e.g., Le Roux et al. 2007; Le Roux 2012; Eldred and Le Roux 2018). For example, Korn and Danilov (2017) have recently proposed a specific mimetic approach to control the well-known spurious mode occurring in triangular C grids (Wolfram and Fringer 2013). Unstructured grid models have been used for coastal applications for many years and they have now reached the application phase for global applications (e.g., Sidorenko et al. 2015; Petersen et al. 2018, manuscript submitted to *J. Adv.*

*Model. Earth Syst.*). A long-standing concern is that computational cost per nominal grid point is generally much larger than for structured grid models, and this problem is further compounded by the absence of time refinement to locally adjust the time step to the mesh resolution when explicit time stepping is used. The test strategy presented in the next section should provide a way to quantify more rationally the difference in terms of computational costs among existing models.

Recent advances also include the development of hybrid (or generalized) vertical coordinate systems based on arbitrary Lagrangian–Eulerian (ALE) methods. The vertical distribution of Eulerian coordinate levels is predefined, whereas Lagrangian coordinate levels freely evolve with the flow. ALE methods combine the advantages of well-defined (i.e., undistorted) meshes and reduced numerical mixing, and also allow adaptation strategies (e.g., Bleck 2002; Burchard and Beckers 2004; White et al. 2009; Leclair and Madec 2011; Petersen et al. 2015). A difficulty in this case is the rezoning (also known as regridding) phase to maintain the integrity of the grid locally and globally.

A tendency in the design of the oceanic dynamical cores is the extension to the nonhydrostatic (NH) equations. The most widely used approach nowadays in oceanic models is based on the incompressible NH system solved using a pressure correction–projection method that requires the solution of a 3D Poisson equation (Lai et al. 2011; Vitousek and Fringer 2013; Voltzinger and Androsov 2016). Recently, Auclair et al. (2018) have proposed the use of the compressible nonhydrostatic ocean equations with the advantage that no global algebraic system needs to be solved to compute NH pressure anomalies, but with the disadvantage to permit acoustic modes. In this case, a specific numerical procedure is required to maintain acceptable stability of the whole code.

**TEST STRATEGY AND BENCHMARK SUITE.** Given the wide variety of choices that need to be made during the development of dynamical cores and their overall complexity, it is crucial to define evaluation methods to compare the behavior of different models. Such effort has been made over the last decade by the global atmospheric community [Dynamical Core Model Intercomparison Project (DCMIP); Ullrich et al. 2012]. In particular, within DCMIP, a collection of test cases that found broad acceptance in the community has been designed and applied by a large number of modeling groups. The workshop highlighted that in the context of the

oceanic community, existing test cases are scattered in the literature and not always fully documented and reproducible. The few existing examples of such effort (e.g., Ezer et al. 2002; Gerritsen et al. 2008; Ilicak et al. 2012; Soufflet et al. 2016) turned out to provide valuable feedback on the consequences of model formulations. A good test case should be easy to configure with analytical data suitable for all horizontal grids and different vertical coordinates and easy to evaluate while being relevant to test a given component of the dynamical core. The evaluation can be done either via analytical solutions (e.g., Bristeau et al. 2018), numerically converged solutions (provided that all models converge toward the same solution), or more subjectively based on an unambiguous physical understanding of the processes (e.g., Marques et al. 2017). Such a benchmark suite is also useful to motivate communication between modeling groups and also to open room for prospective approaches from applied mathematicians to highlight their effectiveness. Existing initiatives aiming at oceanic model intercomparison based on realistic simulations (e.g., Chassignet et al. 2000; Griffies et al. 2009) are generally too complex to clearly associate the observed differences to particular numerical choices. A way to evaluate numerical models in such complex configurations that could gain ground in the next few years is the uncertainty quantification (e.g., Iskandarani et al. 2016). Such an approach provides tools to characterize modeling and numerical sensitivities.

**CHALLENGES.** Throughout the three days of discussion, different current and future challenges have been identified. Addressing these challenges requires closer collaboration between the modeling groups.

*Multiresolution strategy: Block structured versus unstructured.* On the one hand, unstructured grid models have the ability to allow variable-resolution meshes provided an efficient mesh generation tool (e.g., Engwirda 2017), for example, to adapt the resolution to follow the local Rossby radius (Hallberg 2013; Sein et al. 2017). On the other hand, structured grid models can also locally increase the resolution via nesting techniques (Debreu and Blayo 2008; Warner et al. 2010; Debreu et al. 2012) or quadtree–octree refinement (Popinet and Rickard 2007). One advantage of the nesting approach is to allow the adjustment of the time step and the physical parameters to the local resolutions while unstructured models will need scale-aware parameterizations and a specific procedure for time refinement. Other points to investigate are the impact of variable resolution on propagating waves and the optimal layout to build a multiresolution mesh. A more prospective approach could be the use of an adaptive wavelet method (Kevlahan et al. 2015).

*Energy consistency and resolved/unresolved scales coupling.* Energy consistency is an important aspect for the proper interaction between resolved and parameterized scales (e.g., Burchard 2002; Bachman et al. 2017). However, as soon as a numerical core does not globally conserve energy at a discrete level (e.g., because of monotonicity enforcement, vertical remapping, or some form of upwinding), the identification of energy pathways is difficult and requires an in-depth analysis to close the energy cycle (e.g., Marsaleix et al. 2008; Eden 2016), which can be rather tedious (if not impossible) when advanced high-order numerics are used. An alternative could be to opt for an energy-conserving space and time discretization (e.g., Korn 2017; Eldred et al. 2019) with specific care (based on explicit numerical dissipation in the dynamical core and/or on parameterization of subgrid processes) to avoid instability issues of the existing approaches (Bell et al. 2017). This consistent coupling between dynamical cores and subgrid processes (known as physics–dynamics coupling) is an increasingly important topic for the building of geophysical models in general (Gross et al. 2018).

*Vertical coordinates and spurious numerical mixing.* Spurious mixing (especially spurious dianeutral mixing) is a long-standing issue for oceanic dynamical cores (e.g., Griffies et al. 2000). The use of an ALE vertical coordinate is a way to mitigate this issue. Despite the fact that Gibson et al. (2017) suggest that the vertical component of spurious mixing dominates as horizontal resolution increases, it should not overshadow that many components of dynamical cores can be a source of numerical mixing (e.g., horizontal advection, time stepping, the stabilization of the mode-splitting procedure). There is still a need to better understand the implications of different choices of momentum/tracers advection schemes, rezoning, and remapping procedures on numerical mixing. Idealized test cases and efficient diagnostic tools are important to tackle this issue (e.g., Ilicak et al. 2012; Klingbeil et al. 2014). It would also be instructive to keep investigating the improvement of quasi-Eulerian vertical coordinates (e.g., Berntsen 2011). In this context schemes for the internal pressure gradient and for isoneutral tracer diffusion should also be considered (e.g., Shao et al. 2018).

*Nonhydrostatic pressure contribution.* As discussed earlier, many oceanic dynamical cores have the possibility to account for nonhydrostatic effects. A difficulty is now to clearly identify under which conditions relaxing the hydrostatic assumption is necessary and which resolution is required for proper NH modeling. Another challenge is the possibility to locally account for NH effects within a primitive equations model either in the form of a superparameterization (e.g., Campin et al. 2011) or in the form of two-way nesting between coarse hydrostatic and fine nonhydrostatic meshes (e.g., Blayo and Rousseau 2016). Ultimately, further investigations of the merits and flaws of the incompressible versus compressible NH approaches would be worthwhile, also for global applications (Losch et al. 2004).

*Coupling with other Earth system compartments.* Oceanic dynamical cores are often used as a component of larger coupled model systems. Coupling to several other Earth system compartments is common, such as to surface wave, sea ice, atmospheric, biogeochemical, benthic, and hydrological models. The numerical implementation of such coupling can become an issue (e.g., Lemarié et al. 2015a; Beljaars et al. 2017), especially at high coupling frequency and/ or spatial resolution. More systematic analysis of the coupling stability and consistency using simplified equation sets and the design of simplified coupled test cases must be encouraged.

*Vanishing layers, wetting and drying, and shock-resolving numerics.* An accurate treatment of wetting and drying is essential for coastal simulations as well as for climate simulations of under-ice-shelf cavities. At a numerical level this requires the nonnegativity of the water height and an adequate volume-conserving treatment of dry states (also known as vacuum states). Standard numerical methods used in oceanic dynamical cores do not have the ability to handle vacuum, or equivalently shocks. Instead, approaches based on some predefined minimum water depth and specific ad hoc manipulation of discrete fluxes are often used (e.g., section 5.2 in Klingbeil et al. 2018). However, there is a vast literature dedicated to the design of numerical schemes preserving positivity and able to correctly treat vacuum states that furthermore satisfy an entropy-preserving property; that is, the nonlinear solution is physically relevant even in the presence of discontinuities (e.g., Audusse et al. 2004, 2016). Considering advection–diffusion equations, for example, for tracers (temperature, salinity), there is an obvious benefit in using numerical schemes preserving the maximum

principle. There could be an interest in comparing these more advanced approaches with the usual treatment adopted in dynamical cores.

**CONCLUSIONS.** The workshop gave a broad and fresh overview of existing numerical methods used in realistic ocean models as well as some examples of alternatives from the applied maths community. The participants have been enthusiastic and very positive about the possibility to sustain this type of workshop into a biennial workshop series. A collective article is currently in preparation to summarize the challenges and prospects for oceanic numerical cores across all scales. Moreover, a particular effort will be directed toward the formation of an active community of model developers with international collaborations, starting, for example, with the standardization of existing idealized test cases as the basis for model/ methods intercomparison studies. The next meeting will be organized either in fall 2019 or winter 2020 in Hamburg, Germany.

**ACKNOWLEDGMENTS.** The authors thank Gurvan Madec for reading and checking the content of this summary as well as all of the workshop's participants (https://commodore2018.sciencesconf.org/resource /listeparticipants). The workshop was hosted by the French National Institute for computer science and applied mathematics (Inria), Paris, France. We gratefully acknowledge the support from Inria, from the Coastal and Regional Ocean Community Model research group (Gdr Croco), and from the French program Earth's Fluid Envelopes (LEFE).

**REFERENCES**

Auclair, F., L. Bordois, Y. Dossmann, T. Duhaut, A. Paci, C. Ulses, and C. Nguyen, 2018: A non-hydrostatic non-Boussinesq algorithm for free-surface ocean modelling. *Ocean Modell.*, **132**, 12–29, https://doi .org/10.1016/j.ocemod.2018.07.011.

Audusse, E., F. Bouchut, M.-O. Bristeau, R. Klein, and B. Perthame, 2004: A fast and stable well-balanced scheme with hydrostatic reconstruction for shallow water flows. *SIAM J. Sci. Comput.*, **25**, 2050–2065, https://doi.org/10.1137/S1064827503431090.

——, ——, ——, and J. Sainte-Marie, 2016: Kinetic entropy inequality and hydrostatic reconstruction scheme for the Saint-Venant system. *Math. Comp.*, **85**, 2815–2837, https://doi.org/10.1090/mcom/3099.

Bachman, S. D., B. Fox-Kemper, and B. Pearson, 2017: A scale-aware subgrid model for quasi-geostrophic turbulence. *J. Geophys. Res. Oceans*, **122**, 1529–1554, https://doi.org/10.1002/2016JC012265.

Beljaars, A., E. Dutra, G. Balsamo, and F. Lemarié, 2017: On the numerical stability of surface–atmosphere coupling in weather and climate models. *Geosci. Model Dev.*, **10**, 977–989, https://doi.org/10.5194/gmd-10-977-2017.

Bell, M. J., P. S. Peixoto, and J. Thuburn, 2017: Numerical instabilities of vector-invariant momentum equations on rectangular C-grids. *Quart. J. Roy. Meteor. Soc.*, **143**, 563–581, https://doi.org/10.1002/qj.2950.

Berntsen, J., 2011: A perfectly balanced method for estimating the internal pressure gradients in σ-coordinate ocean models. *Ocean Modell.*, **38**, 85–95, https://doi.org/10.1016/j.ocemod.2011.02.006.

Blayo, E., and A. Rousseau, 2016: About interface conditions for coupling hydrostatic and nonhydrostatic Navier-Stokes flows. *Discrete Contin. Dyn. Syst. Ser. S*, **9**, 1565–1574, https://doi.org/10.3934/dcdss.2016063.

Bleck, R., 2002: An oceanic general circulation model framed in hybrid isopycnic-Cartesian coordinates. *Ocean Modell.*, **4**, 55–88, https://doi.org/10.1016/S1463-5003(01)00012-9.

Bristeau, M.-O., B. D. I. Martino, A. Mangeney, J. Sainte-Marie, and F. Souillé, 2018: Various analytical solutions for the incompressible Euler and Navier-Stokes systems with free surface. HAL, 26 pp., https://hal.archives-ouvertes.fr/hal-01831622.

Burchard, H., 2002: Energy-conserving discretisation of turbulent shear and buoyancy production. *Ocean Modell.*, **4**, 347–361, https://doi.org/10.1016/S1463-5003(02)00009-4.

——, and J.-M. Beckers, 2004: Non-uniform adaptive vertical grids in one-dimensional numerical ocean models. *Ocean Modell.*, **6**, 51–81, https://doi.org/10.1016/S1463-5003(02)00060-4.

Campin, J.-M., C. Hill, H. Jones, and J. Marshall, 2011: Super-parameterization in ocean modeling: Application to deep convection. *Ocean Modell.*, **36**, 90–101, https://doi.org/10.1016/j.ocemod.2010.10.003.

Chassignet, E. P., and Coauthors, 2000: DAMÉE-NAB: The base experiments. *Dyn. Atmos. Oceans*, **32**, 155–183, https://doi.org/10.1016/S0377-0265(00)00046-4.

Chen, C., H. Liu, and R. C. Beardsley, 2003: An unstructured grid, finite-volume, three-dimensional, primitive equations ocean model: Application to coastal ocean and estuaries. *J. Atmos. Oceanic Technol.*, **20**, 159–186, https://doi.org/10.1175/1520-0426(2003)020<0159:AUGFVT>2.0.CO;2.

Danilov, S., D. Sidorenko, Q. Wang, and T. Jung, 2017: The Finite-volumE Sea ice–Ocean Model (FESOM2). *Geosci. Model Dev.*, **10**, 765–789, https://doi.org/10.5194/gmd-10-765-2017.

Debreu, L., and E. Blayo, 2008: Two-way embedding algorithms: A review. *Ocean Dyn.*, **58**, 415–428, https://doi.org/10.1007/s10236-008-0150-9.

——, P. Marchesiello, P. Penven, and G. Cambon, 2012: Two-way nesting in split-explicit ocean models: Algorithms, implementation and validation. *Ocean Modell.*, **49-50**, 1–21, https://doi.org/10.1016/j.ocemod.2012.03.003.

Eden, C., 2016: Closing the energy cycle in an ocean model. *Ocean Modell.*, **101**, 30–42, https://doi.org/10.1016/j.ocemod.2016.02.005.

Eldred, C., and D. Y. Le Roux, 2018: Dispersion analysis of compatible Galerkin schemes for the 1D shallow water model. *J. Comput. Phys.*, **371**, 779–800, https://doi.org/10.1016/j.jcp.2018.06.007.

——, T. Dubos, and E. Kritsikis, 2019: A quasi-Hamiltonian discretization of the thermal shallow water equations. *J. Comput. Phys.*, **379**, 1–31, https://doi.org/10.1016/j.jcp.2018.10.038.

Engwirda, D., 2017: JIGSAW-GEO (1.0): Locally orthogonal staggered unstructured grid generation for general circulation modelling on the sphere. *Geosci. Model Dev.*, **10**, 2117–2140, https://doi.org/10.5194/gmd-10-2117-2017.

Ezer, T., H. Arango, and A. F. Shchepetkin, 2002: Developments in terrain-following ocean models: Intercomparisons of numerical aspects. *Ocean Modell.*, **4**, 249–267, https://doi.org/10.1016/S1463-5003(02)00003-3.

Gerritsen, H., E. D. Goede, F. Platzek, J. A. T. M. van Kester, M. Genseberger, and R. Uittenbogaard, 2008: Validation document Delft3D-FLOW. Tech. Rep. X0356/M3470, Deltares, 182 pp., https://oss.deltares.nl/c/document_library/get_file?uuid=39169f8f-4ab0-4f7b-9771-c3f7d0ddd61f&groupId=183920.

Gibson, A. H., A. M. Hogg, A. E. Kiss, C. J. Shakespeare, and A. Adcroft, 2017: Attribution of horizontal and vertical contributions to spurious mixing in an arbitrary Lagrangian–Eulerian ocean model. *Ocean Modell.*, **119**, 45–56, https://doi.org/10.1016/j.ocemod.2017.09.008.

Griffies, S., and A. Adcroft, 2008: Formulating the equations of ocean models. *Ocean Modeling in an Eddying Regime*, *Geophys. Monogr.*, Vol. 177, Amer. Geophys. Union, 281–317.

——, R. C. Pacanowski, and R. W. Hallberg, 2000: Spurious diapycnal mixing associated with advection in a *z*-coordinate ocean model. *Mon. Wea. Rev.*, **128**, 538–564, https://doi.org/10.1175/1520-0493(2000)128<0538:SDMAWA>2.0.CO;2.

——, and Coauthors, 2009: Coordinated Ocean-Ice Reference Experiments (COREs). *Ocean Modell.*, **26**, 1–46, https://doi.org/10.1016/j.ocemod.2008.08.007.

Gross, M., and Coauthors, 2018: Physics–dynamics coupling in weather, climate and Earth system models: Challenges and recent progress. *Mon. Weather Rev.*, **146**, 3505–3544, https://doi.org/10.1175/MWR-D-17-0345.1.

Hallberg, R., 2013: Using a resolution function to regulate parameterizations of oceanic mesoscale eddy effects. *Ocean Modell.*, **72**, 92–103, https://doi.org/10.1016/j.ocemod.2013.08.007.

Ilicak, M., A. J. Adcroft, S. M. Griffies, and R. W. Hallberg, 2012: Spurious dianeutral mixing and the role of momentum closure. *Ocean Modell.*, **45–46**, 37–58, https://doi.org/10.1016/j.ocemod.2011.10.003.

Iskandarani, M., S. Wang, A. Srinivasan, W. C. Thacker, J. Winokur, and O. M. Knio, 2016: An overview of uncertainty quantification techniques with application to oceanic and oil-spill simulations. *J. Geophys. Res. Oceans*, **121**, 2789–2808, https://doi.org/10.1002/2015JC011366.

Kärnä, T., S. C. Kramer, L. Mitchell, D. A. Ham, M. D. Piggott, and A. M. Baptista, 2018: Thetis coastal ocean model: Discontinuous Galerkin discretization for the three-dimensional hydrostatic equations. *Geosci. Model Dev.*, **11**, 4359–4382, https://doi.org/10.5194/gmd-11-4359-2018.

Kevlahan, N. K.-R., T. Dubos, and M. Aechtner, 2015: Adaptive wavelet simulation of global ocean dynamics using a new Brinkman volume penalization. *Geosci. Model Dev.*, **8**, 3891–3909, https://doi.org/10.5194/gmd-8-3891-2015.

Klingbeil, K., M. Mohammadi-Aragh, U. Gräwe, and H. Burchard, 2014: Quantification of spurious dissipation and mixing—Discrete variance decay in a finite-volume framework. *Ocean Modell.*, **81**, 49–64, https://doi.org/10.1016/j.ocemod.2014.06.001.

——, F. Lemarié, L. Debreu, and H. Burchard, 2018: The numerics of hydrostatic structured-grid coastal ocean models: State of the art and future perspectives. *Ocean Modell.*, **125**, 80–105, https://doi.org/10.1016/j.ocemod.2018.01.007.

Korn, P., 2017: Formulation of an unstructured grid model for global ocean dynamics. *J. Comput. Phys.*, **339**, 525–552, https://doi.org/10.1016/j.jcp.2017.03.009.

——, and S. Danilov, 2017: Elementary dispersion analysis of some mimetic discretizations on triangular C-grids. *J. Comput. Phys.*, **330**, 156–172, https://doi.org/10.1016/j.jcp.2016.10.059.

Lai, Z., C. Chen, G. W. Cowles, and R. C. Beardsley, 2011: A nonhydrostatic version of FVCOM: 1. Validation experiments. *J. Geophys. Res.*, **115**, C11010, https://doi.org/10.1029/2009JC005525.

Le Roux, D., 2012: Spurious inertial oscillations in shallow-water models. *J. Comput. Phys.*, **231**, 7959–7987, https://doi.org/10.1016/j.jcp.2012.04.052.

——, V. Rostand, and B. Pouliot, 2007: Analysis of numerically induced oscillations in 2D finite-element shallow-water models Part I: Inertia–gravity waves. *SIAM J. Sci. Comput.*, **29**, 331–360, https://doi.org/10.1137/060650106.

Leclair, M., and G. Madec, 2011: *z*-coordinate, an arbitrary Lagrangian–Eulerian coordinate separating high and low frequency motions. *Ocean Modell.*, **37**, 139–152., https://doi.org/10.1016/j.ocemod.2011.02.001.

Lemarié, F., E. Blayo, and L. Debreu, 2015a: Analysis of ocean-atmosphere coupling algorithms: Consistency and stability. *Procedia Comput. Sci.*, **51**, 2066–2075, https://doi.org/10.1016/j.procs.2015.05.473.

——, L. Debreu, G. Madec, J. Demange, J. Molines, and M. Honnorat, 2015b: Stability constraints for oceanic numerical models: Implications for the formulation of time and space discretizations. *Ocean Modell.*, **92**, 124–148, https://doi.org/10.1016/j.ocemod.2015.06.006.

Losch, M., A. Adcroft, and J.-M. Campin, 2004: How sensitive are coarse general circulation models to fundamental approximations in the equations of motion? *J. Phys. Oceanogr.*, **34**, 306–319, https://doi.org/10.1175/1520-0485(2004)034<0306:HSACGC>2.0.CO;2.

Marques, G. M., M. G. Wells, L. Padman, and T. M. Özgökmen, 2017: Flow splitting in numerical simulations of oceanic dense-water outflows. *Ocean Modell.*, **113**, 66–84, https://doi.org/10.1016/j.ocemod.2017.03.011.

Marsaleix, P., F. Auclair, J. W. Floor, M. J. Herrmann, C. Estournel, I. Pairaud, and C. Ulses, 2008: Energy conservation issues in sigma-coordinate free-surface ocean models. *Ocean Modell.*, **20**, 61–89, https://doi.org/10.1016/j.ocemod.2007.07.005.

Petersen, M. R., D. W. Jacobsen, T. D. Ringler, M. W. Hecht, and M. E. Maltrud, 2015: Evaluation of the arbitrary Lagrangian–Eulerian vertical coordinate method in the MPAS-Ocean model. *Ocean Modell.*, **86**, 93–113, https://doi.org/10.1016/j.ocemod.2014.12.004.

Popinet, S., and G. Rickard, 2007: A tree-based solver for adaptive ocean modelling. *Ocean Modell.*, **16**, 224–249, https://doi.org/10.1016/j.ocemod.2006.10.002.

Ringler, T., J. Thuburn, J. Klemp, and W. Skamarock, 2010: A unified approach to energy conservation and potential vorticity dynamics for arbitrarily-structured C-grids. *J. Comput. Phys.*, **229**, 3065–3090, https://doi.org/10.1016/j.jcp.2009.12.007.

Sein, D. V., and Coauthors, 2017: Ocean modeling on a mesh with resolution following the local Rossby radius. *J. Adv. Model. Earth Syst.*, **9**, 2601–2614, https://doi.org/10.1002/2017MS001099.

Shao, A., A. Adcroft, R. Hallberg, and S. Griffies, 2018: Improvements to an extrema-diminishing, density-preserving lateral diffusion algorithm. *2018 Ocean Sciences Meeting*, Portland, OR, Amer. Geophys. Union, Abstract 318301.

Sidorenko, D., and Coauthors, 2015: Towards multi-resolution global climate modeling with ECHAM6–FESOM. Part I: Model formulation and mean climate. *Climate Dyn.*, **44**, 757–780, https://doi.org/10.1007/s00382-014-2290-6.

Soufflet, Y., P. Marchesiello, F. Lemarié, J. Jouanno, X. Capet, L. Debreu, and R. Benshila, 2016: On effective resolution in ocean models. *Ocean Modell.*, **98**, 36–50, https://doi.org/10.1016/j.ocemod.2015.12.004.

Ullrich, P. A., C. Jablonowski, J. Kent, P. H. Lauritzen, R. Nair, and M. A. Taylor, 2012: Dynamical Core Model Intercomparison Project (DCMIP) test case document. UCAR Tech. Doc., 83 pp., http://www-personal.umich.edu/~cjablono/DCMIP-2012_TestCaseDocument_v1.7.pdf.

Vitousek, S., and O. B. Fringer, 2013: Stability and consistency of nonhydrostatic free-surface models using the semi-implicit θ-method. *Int. J. Numer. Methods Fluids*, **72**, 550–582, https://doi.org/10.1002/fld.3755.

Voltzinger, N. E., and A. A. Androsov, 2016: Nonhydrostatic tidal dynamics in the area of a seamount. *Oceanology*, **56**, 491–500, https://doi.org/10.1134/S0001437016030243.

Warner, J. C., W. R. Geyer, and H. G. Arango, 2010: Using a composite grid approach in a complex coastal domain to estimate estuarine residence time. *Comput. Geosci.*, **36**, 921–935, https://doi.org/10.1016/j.cageo.2009.11.008.

White, L., A. Adcroft, and R. Hallberg, 2009: High-order regridding-remapping schemes for continuous isopycnal and generalized coordinates in ocean models. *J. Comput. Phys.*, **228**, 8665–8692, https://doi.org/10.1016/j.jcp.2009.08.016.

Wolfram, P. J., and O. B. Fringer, 2013: Mitigating horizontal divergence "checker-board" oscillations on unstructured triangular C-grids for nonlinear hydrostatic and nonhydrostatic flows. *Ocean Modell.*, **69**, 64–78, https://doi.org/10.1016/j.ocemod.2013.05.007.

Zhang, Y. J., F. Ye, E. V. Stanev, and S. Grashorn, 2016: Seamless cross-scale modeling with schism. *Ocean Modell.*, **102**, 64–81, https://doi.org/10.1016/j.ocemod.2016.05.002.

---

## Author Response (AR1)

Reviewer 1

General Comments

This work is a detailed study of the reproduction of a river plume by some state-of-the-art unstructured mesh models. The problem is introduced with an excellent analysis of the analytical solution and the results show the reproduction of various characteristics of the plume by the models. The use of different numerical schemes and other model parameters is discussed. The paper presents a considerable amount of work of very good quality and of great interest. However, although the first part is well written, the second part with results and discussion needs a substantial revision. Below are the main comments, which refer from section 5 onwards.

Dear Reviewer 1,

Thank you for your effort to review the paper and your insightful comments!

- This part of the paper is written as a technical report. The authors speak to a reader interested in reproducing their experiments. This greatly limits the paper and makes it less useful for readers interested in applying a model in a real situation. In particular, the paper should answer questions such as: what is the best numerical scheme for reproducing a river plume? What are the minimum horizontal and vertical resolutions still good? I would suggest adding a section of conclusions after section 6, which answers these questions, those posed in the Introduction (p.3, r80-83) and discusses the last sentence of the abstract;

  Thank you for the comment! Yes, we agree that the focus was largely on the reproduction of the test case, which still remains the main goal. However, in the revised manuscript we tried to avoid too technical formulations of the results, proposing a broader view on them. In the revised manuscript, we summarised our findings in the Conclusion Section as recommended. The manuscript aims at assessing the level of numerical mixing for different numerical schemes used in our runs, and also on documenting the test case. However, we cannot state which scheme would perform best in realistic cases, because many other factors may contribute in addition to numerical mixing. This is a much broader topic. We are, of course, interested in it and plan to address it in the future. We did not concentrate on the question about minimum/optimal horizontal/vertical resolution; to answer it, a new series of sensitivity runs should be presented and discussed, which is once again a subject of future work. However, we did some additional experiments: we fixed the depth to 10 meters everywhere (in the presented setups there is a slope, maximum depth is 30 m). This step provides increased resolution and supposes the usage of the layers with nearly constant depths. In this case all solutions are much closer to the analytical one. However, the test case is designed to stress the difference in the performance rather than to find the conditions when the analytical solution can be reached.

We have now added a Conclusion Section, which concentrates on the questions raised in the Introduction and in the last sentence of the Abstract:

[revised manuscript text omitted]

- The part of the results is too long and difficult to read, it should be reduced where possible. Furthermore, after section 4, the English must be carefully checked and improved (you could

contact a native English speaker), trying to use shorter sentences, better use of punctuation and to extend the explanation of some parts with complex concepts which, sometimes, are sketched out;

Thank you for the comment. We went through the Results section, removed some parts and rewrote the others. Also, the proofreading of the second part of the manuscript has been done according to the suggestion.

- I would move section 7 to the appendix, trying to use some tables. I would finish the paper, in a more traditional way, with the Conclusions.

We put Section 7 into Appendix. The table with final characteristics is now in the Conclusion Section.

Specific Comments

- Throughout the paper, references should be made to the numbers of the sections, not to their name;

Done.

- Table 1 would be more convenient at the beginning of section 5, where it is cited many times;

Thank you, we put in the beginning of Section 5.

- From section 5 the Authors use "second (first) inertial period", which is a bit misleading. I would use "two (one) inertial periods" or "two rotational periods", in accordance with the first part of the paper;

We have replaced first/second by one/two. Only in places, where there are 'within' or 'over', we use 'second' or 'first'.

- I think that the comparison with analytical results and laboratory studies should be used more, both in the text and in the figures. In the figures, it would be useful to see these quantities. In any case, I leave the decision to the authors;

In the text we emphasize the comparison to analytical solution more.  We also added Table 3 (see previous answers).

- The figures with the vertical profiles have the x-axis inverted. I find this unintuitive; anyway, it is not so important;

Thank you, we have modified all figures accordingly.

- p21r465-467: Explain more;

In the revised manuscript we have deleted this piece of text.

- Fig. 6: Explain the various panels more. A line Fr = 1 would be useful;

Done.

- p23r478-479: Explain more;

Thank you. Done:

*The ratio between the length (along-shelf spread) and width (offshore spread) of the bulge called ellipticity (Avicola and Huq, 2003) is another parameter, which indicates the presence of numerical mixing in the system. Generally, numerical mixing tends to reduce the bulge external radius due to a decreasing salinity gradient (horizontal, vertical or both) in the near-field or bulge zone and the resulting reduction in plume-associated offshore velocities. Numerical mixing leads to a deepening of the bulge or/and to a changed angle of impingement, such that the center of the bulge gets closer to the coast: the bulge ends up being sliced off by the coastal wall. Numerical mixing therefore tends to increase the ellipticity. It thus comes at no surprise that in all triangular-mesh configurations, including run1, the ratio is too large compared to the expected number (Table 2).*

- p24r495-500: Explain better;

Ok, done:

*The position of the front of the coastal current can also provide a qualitative estimate of the level of numerical diffusion. Numerical mixing moves the bulge center closer to the coast, and hence a larger portion of freshwater enters the coastal current. The position of the head of the coastal current, or the magnitude of its discharge (compared to the analytical solution), can be used to diagnose numerical diffusion in the system (Fig. 5). Note that numerical diffusion levels may be even higher if the same small, offshore restriction of the bulge parallels a weakly developed coastal current. In such a case, the bulge and the coastal current are excessively thick (see e.g., run 6, Fig. 5).*

- Section 5.3: Some parts of the text are missing (p27r516). Another error in r522. The text describes the differences in a concise way, with short comments. Sometimes it is difficult to understand;

Yes, thank you, here is a mistake. Done!

- Section 5.4: Like the previous one, it is hard to understand. Use shorter sentences and describe better the methodology;

Done.

- p29r557: not clear;

We have added an explanation:

*...Among all runs, runs 11 and 12 yield maximum freshwater volumes for the first salinity class. The released freshwater thus largely stays fresh and does not replenish intermediate salinity classes in runs 11 and 12 to the same extent as in the other runs. Runs 1 and 11 suggest that quasi-B discretization on the triangular grid can be a noticeable source of numerical mixing unless noise is suppressed by a filter.*

- p30r563-564: explain better; p30r569: why? Explain more;

Done for both comments:

*It is known (e.g., Hetland, 2005; Burchard, 2020) that a larger isohaline area means less mixing for a given level of freshwater discharge through the isohaline. This fact is illustrated in Equation 16: $f(S) = F(S)/A(S)$. If we keep the diahaline freshwater discharge (**total** transport) $F(S)$ constant and modify the isohaline area, which is situated in the denominator, the respective average diahaline freshwater flux, $f(S)$, would increase. Increased diffusive fluxes in turn mean increased mixing, which in our case is purely numerical. Also the total volume of a particular class can be the same while mixing levels differ from case to case due to different isohaline areas (Eq. 14-16). The volume of the first class can also be larger from one case to another in line with a higher numerical diffusivity level; this would apply, for example, where a plume spreads relatively little and remains relatively thick. To avoid a wrong interpretation of the salinity-volume diagram, one should consider the area of the isohalines (Fig. 10) for each different run. Figure 10 depicts the isohaline areas for different salinity classes. Except for run 7 in Figure 10, which is characterized by the presence of physical eddy diffusivity, all curves have very similar shapes. These shapes reflect that a two-layer system is being considered. The layer occupied by the plume is characterized by low salinities; it is not completely fresh due to the presence of numerical mixing. The plume layer can be characterized by the rapidly growing part of the curve.*

- Fig.11 The curves are different in the panels, use the legends in each panel;

Done.

- p32 605-615 Not clear, write better;

  Yes, thank you. Done:

  *Figure 11b shows the transport through different isohalines: freshwater discharge through different isohalines (Fig. 11a) divided by the corresponding isohaline areas (Fig. 10). Transport naturally tends to decrease as the considered isohaline increases; but even so, the discharge can be the same (see, e.g., run 7 or run 13 in Fig. 10, 11a). Figure 11b sheds some light on vertical near-surface dynamics. For example, Figure 11b shows that run 2 has a large cross-isohaline transport for low salinities. Therefore, despite its generally good performance, this run is characterized by a blurry surface layer (Fig. 3). We also see the principal difference between runs 7 and 13: transport through the freshwater isohaline is largest for 13 but then rapidly decreases, and there is no pronounced mixed layer as there is in run 7.*

- p34r650: compared to? Run 1?

  Compared to run 5. We have re-written the sentences.

- p35r669: First explain the purpose and then describe the runs.

  We have re-written the sentence.

- p36r688: Remove the text in brackets, it is not clear;

  Done.

- p37r690: Where? Explain better where the reader should look;

  Done.

- p38r726: Rephrase the sentence describing your findings not your suggestion.

  Done.

Technical Corrections

- p3r74: the

  Done.

- fig1: r is r_0?

  r is correct.

- p5r131: brackish -> fresh

We have been replaced.

- p5r142: we recommend.. -> we increased ... Describe the set-up, don't give recommendations.

Done.

- p5r146: we suggest -> What did you use? As before, describe the set-up.

Done.

- p7r158: below. Specify the section number.

Done.

- p9r200: a

Done.

- p12r302-305: u_b and h_0 are defined in sec2.1, say it somewhere or remember their definitions;

We have restated their meanings in the text

- p14r357: angel…

Fixed:)

- p21r464: typo

Done.

- p23r490: there

Done.

- Fig. 7: surface v-component?

Yes!, done, thank you.

- p24r505: Fig3, which panel?

We have specified the panel - Fig 3d.

- p28r547: intervening? I don't understand this sentence;

We rephrased the sentence, thank you.

- p23r550:.) ->).

  Done.

- p30r559 signalizing?

  It is rephrased.

- p32r591 than

  'Than' is removed.

- p39r752 general

  Thank you, removed.

Reviewer 2

Authors have done a lot of experiments, and some of them should be useful. While some of them duplicated with previous plume modeling, such as Xia's Cape Fear River Modeling (North Carolina State University, 2007) to discuss how the numerical scheme to impact plume's structure modeling, which is the key part of this manuscript. The horizontal resolution and vertical layers were given discussion in Xia et al., 2007 and other papers which were listed below. Numerical mixing can be considered as numerical errors, and authors need be carefully to utilize the error for the physical explanation. Please re-format this work and do a more than major revision to provide insight the plume community. Please remove most experiments which was conducted from these papers below and other literatures, make this manuscript concise and useful. I will provide more comments after new version, not this long, wordy unclear one. Also did authors simulate the internal wave (line 68?)

Niu, Q., Xia, M. (2021) "The behavior and wind-driven dispersions of two dynamically distinctive limnetic river plumes in a semi-enclosed basin," Estuarine, Coastal and Shelf Sciences.(In press)

Niu, Q., Xia, M., Ludsin, S.A., Chu, P.Y., Mason, D.M., Rutherford, E.S. (2018). "High‐urbidity events in Western Lake Erie during ice-free cycles: Contributions of river-loaded vs. resuspended sediments," Limnology and Oceanography,00, 1-18.

Jiang, L., & Xia, M. (2016). "Dynamics of the Chesapeake Bay outflow plume: Realistic plume simulations and its seasonal, interannual variability," Journal of Geophysical Research: Oceans, 121, 1424-1445.

Xia, M., Xie, L., Pietrafesa, L.J., Whitney, M.M. (2011). "The ideal response of a Gulf of Mexico estuary plume to wind forcing: Its connection with salt flux and a Lagrangian view," Journal of Geophysical Research, 116, C08035.

Xia, M., Xie, L., Pietrafesa, L.J. (2010). "Winds and the orientation of a coastal plane estuary plume," Geophysical Research Letters,37, L19601.

Xia, M., Xie, L., Pietrafesa, L.J. (2007). "Modeling of the Cape Fear River estuary plume," Estuaries and Coasts, 30(4), 698-709.

Dear Reviewer 2,

Thank you for your comments. This manuscript deals with a much more basic problem than assumed by the reviewer. It diagnoses numerical diffusion in coastal models, showing that it is related to advection schemes rather than to discretization or mesh type. There are many other important factors that influence plume propagation in realistic configurations, as explored in the papers cited by the reviewer, but all they work on the top of model numerical factors. We surely agree that the questions raised in the papers cited by the reviewer are important in real-world applications, that many important aspects of plume dynamics were carefully explored in the papers cited by the reviewer. However, the level of numerical diffusion is also important,

which is sometimes not fully appreciated. The present manuscript is addressing this question using a specially designed plume configuration that does not characterize a particular river.

It is important to note that all simulations considered in the manuscript

1. do not consider the wind, wave or tidal forcing as all of the listed by Reviewer 2 articles;

2. are done without physical eddy diffusivity (except one to see the difference) to be able to trace the level of numerical mixing accurately;

3. are performed by different models using different schemes.

So, we present a novel test case and diagnostic metrics by which the numerical mixing in ocean models can be quantified. Such a test case can be rather helpful for model developers, especially as it allows comparing models with very different numerical discretizations (in the present paper we consider finite volume C and quasi-B grid models on unstructured and structured meshes, as well as a discontinuous Galerkin finite element model on an unstructured mesh). We believe that the novel test case and metrics are of interest to the model development community.

The references proposed by the reviewer are all examples of excellent work, but they go beyond the present study as they deal with processes not considered by us (e.g. impact of winds, river discharge, or tides) or focus on model assessment in realistic river plume simulations. They do not focus on the systematic assessment of numerical mixing as the present work does. We also note that there are numerous regional studies (some of them are cited in the manuscript), which focus on plume behaviour under different conditions in the different areas worldwide, and we are citing a few of them, including the work proposed by this reviewer.

We have revised the manuscript substantially, shortening and making it easier to read.

I don't think authors have read these references. Clearly Xia 2007 discussed how the numerical schemes impact the plume dynamics, and this submission has overlap with other references. Ideal experiments is very simple, and plume should be investigated with realistic as well. Under the strong river runoff, most mixing scheme won't work well

Dear Reviewer 2,

Suggested reference to Xia et al. 2007 "Modeling of the Cape Fear River estuary plume" does not analyze performance of different tracer advection schemes or limiters with respect to numerical mixing. The individual and coupled effects of the astronomical tides, river discharge, and atmospheric winds were considered to investigate the Cape Fear River Estuary dynamics. On page 699 (right side) the paper provides a brief description of the used EFDC model. Only here the advection scheme is mentioned : 'The model includes the anti-diffusion upwind

advection scheme that is more suitable for the plume study than the upwind scheme or the central difference scheme (Berdeal et al. 2002). '

It looks like a misunderstanding, because we do not consider different mixing schemes, indeed physical eddy diffusivity is set to 0, eddy viscosity is calculated based on k-eps style turbulence closure or set to background value. We consider the numerical mixing level attributed to the different tracer advection schemes and limiters. This is clearly written through the paper.

As we have mentioned in our first reply, we propose the test case and diagnostic metrics by which the numerical mixing in ocean models can be quantified. Spurious numerical mixing in circulation models can destroy stratification and frontal features, and significantly alter the plume dynamics. While considering the effect of physical forcings on the plume dynamics is certainly relevant, it is out of the scope of the present article. The community needs benchmark test cases (see attached Lemarié et al., 2019 summary ) that are reproducible, can be compared against analytical solutions and offer the analysis of "isolated" effects (not blurred by the interplay of many different processes as in complex realistic scenarios) and the direct connection to specific numerical choices in the model core. The suggested idealized plume scenario with a unique set of parameters is reproduced differently by different, but commonly used, advection schemes+limiters. And this is not surprising because the plume dynamics in some zones can be characterized by nonlinear flow regimes with sharp frontal boundaries. Simplicity or non-simplicity of idealized experiments depend on the accuracy level you chose as acceptable.

We have read all the papers suggested by the Reviewer. However, their scopes are beyond that of our study. Please, find below a brief report on the suggested articles:

1.Niu, Q., Xia, M. (2021) "The behavior and wind-driven dispersions of two dynamically distinctive limnetic river plumes in a semi-enclosed basin," Estuarine, Coastal and Shelf Sciences.(In press)

We did not find an article which exactly matches the title, perhaps the Reviewer meant 'The behaviors of two limnetic river plumes discharging into the semi-enclosed western basin of Lake Erie during ice-free seasons' .

The article is about wind-driven dynamics of the Detroit and Maumee River sediment plumes in the semi-enclosed western basin of Lake Erie on several temporal scales. In our case study the wind- driven dynamics has not been considered.

2. Niu, Q., Xia, M., Ludsin, S.A., Chu, P.Y., Mason, D.M., Rutherford, E.S. (2018). "High turbidity events in Western Lake Erie during ice-free cycles: Contributions of river-loaded vs. resuspended sediments," Limnology and Oceanography,00, 1-18.

The article investigates the contributions of river loading (Detroit and Maumee Rivers) versus resuspension to high-turbidity events in Western Lake Erie during ice-free conditions in 2002–2012 using a wave-current forced sediment model (FVCOM based). The major result is that suspended sediment dynamics and high turbidity events in the area were dominated by wind and waves in the offshore regions, and were driven by river loadings near the mouths.

We agree that it is a very important regional study, however, it has focus on sediment dynamics and is hardly relevant to our idealised plume scenario and its major aim.

3. Jiang, L., & Xia, M. (2016). "Dynamics of the Chesapeake Bay outflow plume: Realistic plume simulations and its seasonal, interannual variability," Journal of Geophysical Research: Oceans, 121, 1424-1445.

The article identifies five types of real-time plume behavior regulated by wind and river discharge. Also it contains some sensitivity experiments related to the grid cell sizes considering fine and coarse grids. The article gives very valuable insights about Chesapeake Bay outflow plume behaviour. However, these five types are defined based on presenting physical conditions (preliminary by wind conditions) and there is no established connection to the numerical scheme performance. Therefore, the topic and analysis of the paper are beyond the topic and aims of our manuscript.

4. "The ideal response of a Gulf of Mexico estuary plume to wind forcing: Its connection with salt flux and a Lagrangian view," Journal of Geophysical Research, 116, C08035.

The questions posted by the article are:

1) How does wind forcing affect bay water as it encounters the Gulf?

2) How do plume distribution, fluxes, and particle transport change with changing wind conditions?

The questions are important to understand the regional dynamics. However, as they deal with physical forcings, the topic is beyond the current study.

5.Xia, M., Xie, L., Pietrafesa, L.J. (2010). "Winds and the orientation of a coastal plane estuary plume," Geophysical Research Letters,37, L19601.

The suggested article deals with the Cape Fear River Estuary. and its river plume behavior (type) under different wind forcing and river discharge conditions. Results showed that wind direction, wind speed, and to a lesser extent river discharge contribute to plume transitions from one type to another among six defined major types. This topic is interesting, but not relevant for our study.

6. Xia, M., Xie, L., Pietrafesa, L.J. (2007). "Modeling of the Cape Fear River estuary plume," Estuaries and Coasts, 30(4), 698-709.

Please, see the comment above.